# THE COST OF DOWN-SCALING LANGUAGE MODELS: FACT RECALL DETERIORATES BEFORE IN-CONTEXT LEARNING

**Tian Jin**[*1]   **Nolan Clement**[*2]   **Xin Dong**[*3]
**Vaishnavh Nagarajan**[4]   **Michael Carbin**[1]   **Jonathan Ragan-Kelley**[1]
**Gintare Karolina Dziugaite**[5]

[1]MIT CSAIL   [2]MIT   [3]Harvard University   [4]Google Research   [5]Google DeepMind

## ABSTRACT

How does scaling the number of parameters in large language models (LLMs) affect their core capabilities? We study two scaling techniques: weight pruning and *dense scaling*, which is simply training a smaller or larger model. Our study focuses on how these techniques affect two core capabilities of LLMs: (a) recalling facts presented during pre-training and (b) processing information presented in-context during inference. By curating a suite of tasks that help disentangle these two capabilities, we find a striking difference in how these two capabilities evolve due to scaling. Reducing the model size by more than 30% (via either scaling approach) significantly decreases the ability to (a) recall facts seen in pre-training, leading to more than 5% drop in relative accuracy. Yet, a 60–70% reduction largely preserves the various ways the model can (b) process in-context information, ranging from retrieving answers from a long context to learning parameterized functions from in-context examples, causing less than 5% drop in relative accuracy. The fact that both dense scaling and weight pruning exhibit this behavior suggests that scaling model size has an inherently disparate effect on fact recall and in-context learning.

## 1 INTRODUCTION

Scaling up the size of large language models (LLMs) has yielded impressive performance gains on many natural language tasks (Kaplan et al., 2020; Hoffmann et al., 2022; Brown et al., 2020; Wei et al., 2022). On the other hand, to deploy language models sustainably, it is also critical to scale them *down* while preserving their end utility. Naturally, there is a growing interest in both scaling up and scaling down language models (Kaplan et al. (2020); Hoffmann et al. (2022); Frantar & Alistarh (2023); Jiang et al. (2023); Kurtic et al. (2022); Santacroce et al. (2023)). Most of these works evaluate size–performance trade-offs of scaling through aggregate performance metrics such as perplexity or downstream accuracy on existing benchmarks.

However, we argue that there exist subtle but important effects of scaling that standard metrics alone cannot capture. Earlier works on image models found, for example, that pruning can bias the model against certain label categories (Hooker et al., 2021) or improve robustness against noisy training data (Jin et al., 2022) — these effects do not simply manifest in the overall accuracy of the model. Taking inspiration from these works, we identify the subtle effects of scaling LLMs (up or down) on their *capabilities*, the ability to perform fundamental tasks that underpin many applications.

**Our approach.** In this work, we study the effects of scaling the number of parameters in an LLM on two fundamentally dichotomous capabilities put forth by Chan et al. (2022b;a): *fact recall* — the capability to process information stored in weights (such as facts seen during pre-training) and *in-context learning* — the capability to process information stored in context (such as hints given in the prompt). It is, however, challenging to isolate the evaluation of these capabilities by simply measuring performance on arbitrary downstream tasks — after all, most tasks require both capabilities

---

[*]Correspondence to: `tianjin@csail.mit.edu`, `mcarbin@csail.mit.edu`, `gkdz@google.com`. .

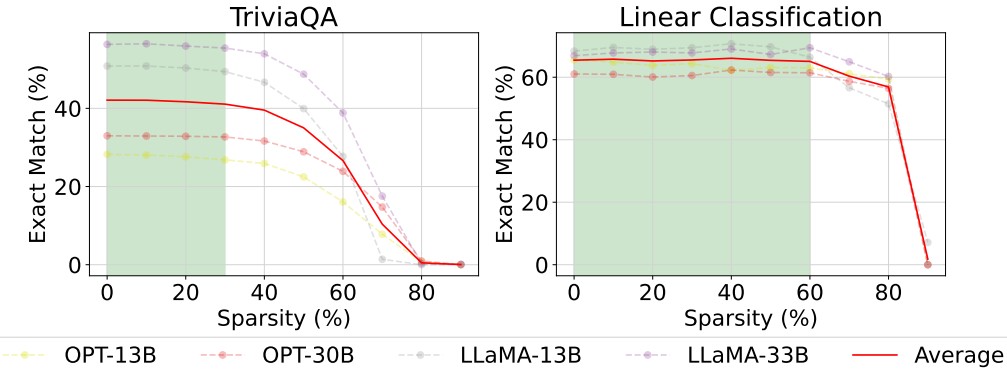

Figure 1: Pruning to moderate sparsity ($> 30\%$ sparse) harms fact recall while in-context learning withstands even aggressive pruning (60% sparse). We plot the accuracy versus sparsity for the OPT-13B, OPT-30B, LLaMA-13B, and LLaMA-33B models on TriviaQA dataset for fact recall evaluation (left) and linear classification dataset for ICL evaluation (right). We plot their average performance in red. We color the range of sparsity where accuracy drop is within (relative) 5% of the dense model in green. Accepting (relative) 5% accuracy drop w.r.t. the dense model, the maximum sparsity on fact recall task is 30%, whereas the maximum sparsity on ICL task is 60%.

to varying extents to solve. Therefore, we carefully curate a suite of benchmarks that help tease apart the performance of any given model on the two capabilities. We then evaluate the aforementioned two capabilities under two different scaling techniques: *pruning* and *dense scaling*. For the former, we consider recent scalable pruning methods that remove weights and subsequently update the remaining weights in one shot (Frantar & Alistarh, 2023; Sun et al., 2024); for the latter, we use (separately-trained) dense models with increased or reduced width and depth.

**Benchmarks.** Our curated suite of benchmarks involve four classes of tasks, featuring a *gradation* of reliance on information stored in-weights and in-context. The first three are Q&A tasks: (1) a *closed book Q&A task* that necessitates fact recall from pre-training for which the model needs to recall information from the weights, (2) an *open book Q&A task* for which it suffices to copy information from the context and (3) an *overriding Q&A task*, where the information provided in-context overrides facts seen during pre-training (Li et al., 2023; Longpre et al., 2021). Finally, to evaluate more sophisticated in-context learning capabilities, we test the model on (4) *learning tasks*, where the model must learn parameterized classification functions (e.g., linear classifiers) using input–output pairs presented in-context. Overall, we consider close-book, overriding Q&A and learning tasks as a sequence of tasks that requires increasing levels of in-context learning (ICL) capabilities (See Section 3 for more discussion). We summarize these tasks in Table 1.

**Findings.** We find surprisingly dichotomous effects of down-scaling on fact recall and ICL capabilities. For fact recall, minimal pruning significantly degrades performance (Section 4) – specifically, for the task in our benchmark that relies most heavily on the fact recall capability, removing more than 30% of weights leads to significant ($> 5\%$, relative) accuracy degradation (Figure 1, left). This is surprising, given that prior pruning literature demonstrates significant down-scaling without noticeable change to accuracy: pioneering work (Dong et al., 2017) removes $\sim 45\%$ of the weights in a ResNet50 model without noticeable accuracy degradation. [1] In contrast, ICL withstands significant pruning (Section 5). Specifically, for the task that relies most heavily on the ICL capability, even after pruning 60% of weights, the relative accuracy decrease is less than 5% (Figure 1, right). Our results challenges the hypothesis presented in the seminal work of Brown et al. (2020), that scaling up the number of parameters strongly benefits the ICL capability of LLMs.[2] Furthermore, fact recall and ICL shows the same sensitivity and robustness to dense scaling, respectively (Section 6). Our work reveals the disparate effects of scaling on LLM capabilities, which notably holds for two substantially different types of scaling techniques: pruning and dense scaling. We hope that researchers and practitioners may apply our findings to:

---

[1] Please refer to Section 4.1, Figure 2.(a) of Dong et al. (2017). Notably, Dong et al. (2017) obtained these results similarly by pruning without retraining.

[2] "Since in-context learning involves absorbing many skills and tasks within the parameters of the model, it is plausible that in-context learning abilities might show similarly strong gains with scale." (Brown et al., 2020)

Table 1: Tasks employed in our experiments.

| Task Type | Task | Context Type | Recalling Facts from Pre-training | Retrieving Facts from Contexts | Learning Patterns from Contexts | Considered ICL |
|---|---|---|---|---|---|---|
| Q&A | WebQuestions | Empty | ✓ | ✗ | - | ✗ |
| | NaturalQuestions | Factual | ✓ | ✓ | - | ✓ |
| | TriviaQA | Empty | ✓ | ✗ | - | ✗ |
| | TriviaQA (Filtered) | Factual | ✓ | ✓ | - | ✓ |
| | DisentQA | Overriding | ✗ | ✓ | - | ✓ |
| Learning | Linear CLS | Examples | - | - | ✓ | ✓ |
| | 2-layer NN | Examples | - | - | ✓ | ✓ |
| | Decision Tree | Examples | - | - | ✓ | ✓ |

*Improve inference efficiency.* Our work reveals that scaling down model size alone has little impact on tasks demanding processing information in the LLM's context. Practitioners may thus use our findings to identify scenarios where decisions could be routed to a smaller model instead of a larger one without compromising task performance (Chen et al., 2023; Dohan et al., 2022).

*Improve assessment of down-scaling techniques.* Whether through pruning or training a distinct, smaller model, our work calls for a balanced evaluation of the down-scaled model prior to its adoption. Overemphasis on either fact recall or ICL risks skewing the assessment of LLM capabilities.

*Improve LLM interpretability.* We find a remarkably small set of weights responsible for ICL. This underscores the potential of pruning as a tool for isolating neurons responsible for LLM capabilities.

**Contributions.**

1. We curate a set of benchmarks for assessing the disparate effects of down-scaling on core capabilities of LLMs, focusing on fact recall and in-context learning (ICL).

2. Using these benchmarks, we evaluate two scalable weight pruning algorithms for LLMs. We investigate an extensive set of models comprising six base LLMs, with sizes reaching up to 33 billion parameters. We find that:

   - even moderate levels of pruning ($> 30\%$) hurt fact recall. However, when the evidence required to solve a question-answering task is provided in context, the model's ability to correctly answer questions survives to higher levels of sparsity;
   - in contrast, ICL withstands aggressive pruning (up to 60–70%).

3. Similarly for dense scaling, we find the same disparate patterns as above for fact recall and ICL capabilities as we vary dense model size. This underscores that the disparate effects on ICL and fact recall are not exclusive to pruning but are broader characteristics of scaling in general.

## 2 RELATED WORK

**LLM scaling.** Studies on scaling laws (Hoffmann et al., 2022; Kaplan et al., 2020) suggest a predictable relationship between the quality of language models (i.e., perplexity) and the size of the pre-training corpus and model. Pursuant to this, many have scaled up and discovered remarkable capabilities of language models. Brown et al. (2020) discover that LLMs can perform in-context learning (ICL) effectively: the model learns to perform tasks based on a few examples of input–output demonstrations in the model's context. Other studies (Devlin et al., 2019; Wei et al., 2022; Liang et al., 2023; Srivastava et al., 2023; Gao et al., 2021; Ganguli et al., 2022; Bubeck et al., 2023; Biderman et al., 2023a) evaluate LLMs across many tasks and metrics and assess the utility of scaling. Our work differs in two ways: instead of measuring model performance on arbitrary tasks, we focus on the foundational capabilities of fact recall and ICL. These capabilities underpin the success of many real world applications of LLMs. Furthermore, while prior work (Kaplan et al., 2020) studies joint scaling of both pre-training corpus size and model size, we focus on scaling model size alone.

**In-weight versus in-context learning.** For LLMs, learning occurs both during weight updates and context processing (Chan et al., 2022a). Recent research illustrates the similarities and differences between these learning approaches: Von Oswald et al. (2023) demonstrated that ICL can implement an algorithm akin to gradient descent, commonly associated with in-weight learning. Akyürek et al. (2023) revealed that ICL resembles various in-weight learning algorithms, depending on model size.

Chan et al. (2022b;a) study how the properties of the data distribution disparately affect the two learning approaches. Our work identifies yet another difference: scaling model size has distinct impacts on knowledge learned through model weights and knowledge acquired from model context.

**Neural network pruning.** Pruning removes unimportant parameters in a model. The origin of pruning traces back to LeCun et al. (1990) and Hassibi et al. (1993), with a focus on reducing the computational footprint. More recently, with the advent of deep learning, pruning research has seen a resurgence (Renda et al., 2020; Han et al., 2016; Liu et al., 2017; Frankle et al., 2021; Molchanov et al., 2017; Dong et al., 2017; Ma et al., 2023; Tao et al., 2023; Kurtic et al., 2022; Dettmers et al., 2024). Though pruning traditionally focuses on preserving aggregate metrics such as accuracy, the versatility of LLMs calls for a different approach to assessing pruned models. To this end, Frantar & Alistarh (2023); Sun et al. (2024) assessed model size and task accuracy trade-off. Our work continues to expand our toolkits for empirical assessment, proposing to evaluate pruning's effect on fact recall and ICL. Our investigation extends a growing line of studies on effects of pruning beyond aggregate metrics such as accuracy and perplexity. Hooker et al. (2021; 2020) show that pruning may harm under-represented categories of examples; Liebenwein et al. (2021) suggest that the pruned models are less accurate than the dense one when predicting out-of-distribution examples. Jin et al. (2022) demonstrate that pruning mitigates the harmful effects of noisy data on generalization.

Concurrent to our work, Yin et al. (2023); Jaiswal et al. (2024) remarked on the pronounced negative effect of pruning on knowledge-intensive tasks. These works extensively evaluate tasks that require capabilities of fact recall and in-context information retrieval. Our work additionally curates a gradation of tasks that further *disentangle* these abilities — specifically via overriding QA and ICL tasks, which these concurrent works do not cover.

## 3 PRELIMINARIES

In this section, we introduce methodological details common to all our experiments.

**Pruning algorithms.** We investigate pruning as one possible technique to (down-)scale LLMs. Few pruning algorithms currently scale to LLMs. We use SparseGPT (Frantar & Alistarh, 2023) in the main text and Wanda (Sun et al., 2024) in Appendix D. Both are one-shot pruning algorithms that scale to LLMs and outperform magnitude pruning (i.e., pruning the smallest magnitude weights), without computationally intensive re-training (Frantar & Alistarh, 2023). SparseGPT prunes each layer of the language model by minimizing the $\ell_2$-distance between the outputs of the original dense layer and the pruned layer. It computes these outputs based on a small training dataset. Please refer to Appendix M for implementation details of SparseGPT and Wanda pruning algorithms.

Following standard practice (Frantar & Alistarh, 2023; Frankle & Carbin, 2019), we exclusively prune fully-connected layers. Fully-connected layers contain most of the parameters in attention and feed-forward modules and they account for over 97.5% of the parameters in all models examined. We exclude embedding layers, language modeling heads, and normalization layers from pruning.

**Models.** We evaluate 6 models from 3 families: OPT (Zhang et al., 2022), LLaMA (Touvron et al., 2023) and Pythia (Biderman et al., 2023b). We focus on OPT and LLaMA in our main text and present Pythia results in Appendix E. Pythia family models show consistent results as LLaMA and OPT family models. From the OPT family, we evaluate the two largest models that fit in our hardware setup – OPT-13B and OPT-30B, with 13 and 30 billion parameters, respectively. From the LLaMA family, we evaluate LLaMA-13B and LLaMA-33B, with 13 and 33 billion parameters, respectively.

A notable difference between OPT and LLaMA families is the ratio of pretraining data to model parameters. Zhang et al. (2022) train the OPT models using 180 billion tokens, resulting in ratios of approximately 14 and 5.5 tokens *per* parameter for the two OPT models we consider. Touvron et al. (2023) train the LLaMA-13B model with 1 trillion tokens, resulting in 77 tokens per parameter, and the LLaMA-33B model with 1.4 trillion tokens, resulting in 42 tokens per parameter.

**Benchmark design.** We evaluate models on two complementary capabilities: the capability to call on information seen during pre-training (and then stored in the model's weights) and the capability to call on information presented in-context. Unfortunately, it is difficult to construct tasks that strictly isolate one of these two capability: on the one hand, for every conceivable natural language task, the model must rely on the semantic representations of language it learned from pre-training and

stored in the weights. On the other, every conceivable task necessitates processing model context to understand the instructions in the first place. To help disentangle the two broad capabilities, we focus on two well-defined instantiations of these capabilities: (a) the ability to recall *facts* from pre-training to answer questions and, complementing that (b) the ability to *learn* patterns from context.

We isolate (a) by evaluating the model on recalling facts from pre-training data not provided in-context. We do this using closed-book QA benchmarks.

To evaluate (b), we consider a gradation of tasks that require the increasingly advanced ways of learning from in-context information. Broadly, we consider a series of in-context learning tasks: tasks where the context contains examples of (query, answer) pairs in some form, followed by a test query that the model has to answer.

First, as a simple ICL task, we consider an Open-book QA (Section 4) counterpart of our QA task. In this task setup, the context includes supporting evidence that directly helps answer the question. Essentially, the ICL task involves a test query that appears within the evidence as an in-context example, complete with its correct answer. For instance, if the test query asks, "Who is the author of the novel The Eagle Has Landed?", the supporting evidence would state, "The author of The Eagle Has Landed is...", and include the answer, "Jack Higgins". Note that in addition to the aforementioned evidence–question–answer triplet, we further aid the model with an independent one-shot triplet to demonstrate how to format its answer.

In the above case, the answer provided in-context may also have been present in pre-training data; thus the model may opt to ignore the context, and recall facts from pre-training, and still succeed at the task. Thus, to more rigorously isolate the two mechanisms, we proceed to evaluate on the Overriding QA task (Section 4), where the in-context evidence *contradicts* a fact present in pre-training data. We present a one-shot evidence–question–answer triplet, where the correct answer follows from contextual evidence rather than from pre-training data. Thus, to solve the task correctly, the model must retrieve from the context rather than recall pre-training facts.

Next, we investigate ICL capabilities more sophisticated than *copying* an answer present as an in-context example. To this end, we are interested in tasks where the model must respond to a test query that it has not already seen in-context. A frequently cited example of ICL is the English–French translation task (Brown et al., 2020), where pairs of English and French phrases serve as examples, with the test query being a novel English phrase. However, such conventional instances of ICL tasks require significant background knowledge of languages not available in the context, thereby conflating the evaluation of ICL with the recall of information learned during model pre-training.

To ensure we disentangle the role of knowledge recall from ICL, we consider tasks where the query–answer mapping is a parameterized function (e.g., linear classifiers). Arguably, learning this mapping requires advanced context-processing mechanisms that go beyond information-retrieval mechanisms required for the Open-book and Overriding QA tasks.

**Metrics.** We reproduce the effects of down-scaling on perplexity as in Frantar & Alistarh (2023); Sun et al. (2024) in Appendix A. However, our main metric is *exact match accuracy* (of the answer to each task) because it is a direct measure of the model's ability to satisfactorily perform a task.

**Software and hardware.** We perform pruning on 80GB version of A100 GPUs, using the implementation of SparseGPT (Frantar & Alistarh, 2023). We perform our evaluations using TPU v3 running PyTorch (Paszke et al., 2019) with XLA backend. Our research project consumes approximately 3 TPU-month worth of computation. Using TPUs necessitates bfloat16 numerical format. We assess its impact on our experimental results in Appendix I.

## 4 Effect of Pruning on Question Answering Tasks

The ability of LLMs to answer questions underpins many applications of LLMs. Within a question-answering evaluation framework, we examine the effect of pruning on its capability to recall facts learnt during pre-training. We also test a simple form of ICL capability – extracting answers from supporting evidence in the model's context, where the answer is unlikely available in pre-training data. Finally, we evaluate an interpolation between these two capabilities – extracting answers from supporting evidence in the model's context, where the answer is likely also available in pre-training data. Our results expose that, with supporting evidence in question-answering tasks, the model's

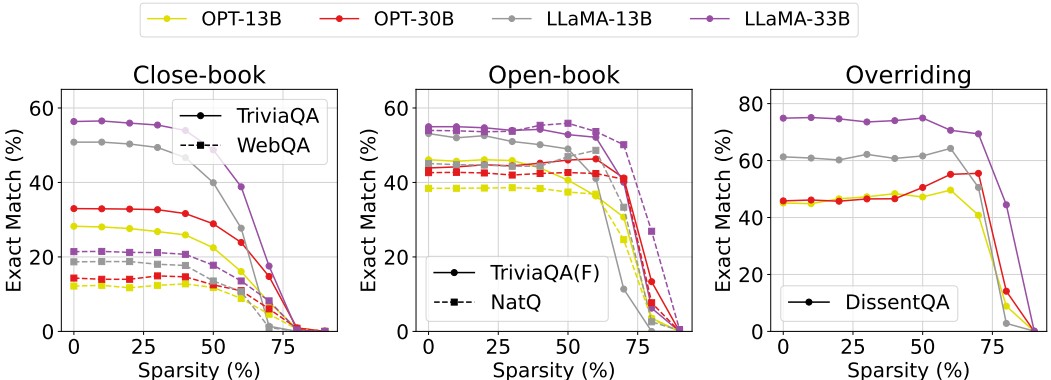

Figure 2: Evaluation for question answering tasks under pruning. Each color represents a different model; each line-style/marker-shape combination represents a different QA dataset. Moderate pruning harms LLMs' capability to recall facts learnt during pre-training: in close-book QA tasks, accepting a 5% drop in average accuracy w.r.t. the dense models, we can prune to 30 and 40% on TriviaQA and WebQA dataset. However, when we present the model with the supporting evidence to answer the question in its context, the model's ability to answer questions survives to higher sparsity: on open-book QA tasks, accepting the same drop in average accuracy, we can prune to 50% and 60% on TriviaQA(F) and NatQ dataset, respectively. On the overriding task (DissentQA dataset), we may even prune to 70% while maintaining the same acceptable relative accuracy drop.

ability to answer questions withstands more aggressive pruning than without. With no supporting evidence in context, accuracy rapidly drops as sparsity increases.

**Datasets.** We use these datasets: (a) *TriviaQA.* Joshi et al. (2017) developed the TriviaQA dataset with questions and supporting evidence. We use its Wikipedia validation partition consisting of 7993 questions. (b) *WebQuestions.* Berant et al. (2013) collected question-answer pairs from the Freebase knowledge database. We use its test set consisting of 2032 questions. (c) *NaturalQuestions.* Kwiatkowski et al. (2019) compiled the NaturalQuestions dataset from Google search queries. We sampled a 7700-question subset of its validation partition (the same size as the following dataset derived from it), to make our evaluation computationally feasible. (d) *DissentQA.* Neeman et al. (2023) constructed the DissentQA dataset from the NaturalQuestions dataset. It contains pairs of questions and evidence for a made-up answer that is different from the factual one. We use its validation partition consisting of 7700 questions.

**Evaluation setup.** Using the aforementioned datasets, we evaluate the ability of pruned models on the following task setup: (i) *Close-book*. We feed the question without any supporting evidence to the model. We use the Wikipedia partition of the TriviaQA dataset and the WebQA dataset for this setup. (ii) *Open-book*. We feed the question with supporting evidence to the model in its context. Notably, this evaluation setup only evaluates whether the answer is right; it is however agnostic to the *mechanism* by which the model retrieves the answer: the model may either generate its answer using the supporting evidence, or by recalling facts from pre-training. To create this dataset, we use a subset of the TriviaQA dataset whose context can fit within the maximum sequence length of the model, consisting of 881 questions. We denote this filtered subset as TriviaQA(F). Additionally, we use the NaturalQuestions dataset for this setup with factual contexts. (iii) *Overriding*. We present the question accompanied by evidence that deviates from the facts presented during pre-training. Given that the anticipated made-up answers are randomized and different from the factual ones, the model is likely unable to rely on memorization from its pre-training data to generate responses. This evaluation rigorously assesses the model's capability to override its pre-training data with new, context-specific information. We use the DisentQA dataset with supporting evidence for the made-up answer for this setup. Examples of the setups are shown in Appendix G.

To summarize: across these setups, our prompts have three parts to them: first, (1) an example evidence–question–answer triplet (or question–answer pair in close-book setting) for demonstration, (2) the supporting evidence for a test question (except in the close-book setup), and (3) a test question to answer. Answers are the model's prompt completions using greedy decoding. We report the percentage of answers that exactly match ground truth.

**Close-book versus open-book results.** Left 2 plots of Figure 2 shows close-book and open-book results. Notably, for all the models, the pruned model maintains performance on open-book tasks until much higher sparsity levels compared to close-book tasks. In particular, when we accept a relative decrease of 5% from pruning in the mean accuracy over four models, the highest achievable sparsities for closed-book tasks are 30% on TriviaQA and 40% on WebQA.

In contrast, maintaining the same acceptable performance drop, the highest acceptable sparsity levels for open-book tasks are 50% on TriviaQA(F) and 60% on NaturalQuestions. Our results suggest that while pruning hurts the model's ability to recall information from its pre-training data at moderate sparsity (30–40%), one possible remedy is to provide the model with supporting evidence in context. The model's ability to answer questions with supporting evidence in its context remains largely intact to higher sparsity levels (50–60%).

**Overriding results.** The rightmost plot of Figure 2 demonstrates a consistent trend for the overriding tasks. On the DisentQA dataset with supporting evidence for the made-up answers, the highest achievable sparsity is 70% allowing for the same 5% accuracy drop as before. Recall that to solve the overriding task, the model must rely on in-context information rather than in-weights information from the pre-training data. Thus, our observations here further substantiate that the model's ability to extracting information from its context remains largely intact at higher sparsity levels.

**Accuracy improvements.** We note that, surprisingly, on open-book and overriding tasks, there can be small accuracy boosts from pruning. For example, the pruned OPT-30B sees a 9.7% accuracy improvement on DissentQA dataset in overriding task setup. The pruned LLaMa-13B sees 3.5% accuracy improvement on NaturalQuestions in the open-book task setup. Though prior work often reports accuracy improvements with pruning (Frankle & Carbin, 2019; Renda et al., 2020), we are, to the best of our knowledge, the first to observe such improvement in the QA setup.

**Takeaways.** Our findings suggest that scaling via pruning has a much higher impact on the model's capability to retrieve facts learnt during pre-training than on its capability to retrieve information from the context. Pruning may even improve model's capability to answer questions when its context includes the necessary information to answer the question.

## 5   Sophisticated In-context Learning

Section 4 demonstrates that moderate pruning preserves question answering task accuracy when supporting evidence is available in context. In this section, we show that even with more sophisticated ICL tasks than previously studied, moderately pruned models maintain their ICL abilities.

Typical ICL tasks require models to have extensive knowledge about language and the world learnt during pre-training. However, Section 4 shows that pruning likely impairs such knowledge. So, to isolate the LLMs' ICL capability from its fact recall capability, we select tasks that require learning a *parameterized* function (e.g., a linear model) through input-output examples present in-context.

**Method.** We evaluate LLMs with multiclass classification problems of the form $f : \mathbb{R}^D \to \{0, 1, \cdots K - 1\}$ from in-context examples, where $D$ refers to the input dimensionality and $K$ the number of label categories. To generate a sequence consisting of a context, query and an answer, we first pick a random parameterized function $f$ from one of three function classes: a). 2-way linear classifiers, b). 2-way neural network classifiers with 2 layers, where the hidden dimension size equals the input dimension size, and c). 4-way decision tree classifiers. For instance, to pick a random 2-way linear classifier, we sample a random classification hyperplane in the input space; see Appendix B for more details regarding the construction of $f$. Then, in the context, we provide $N$ examples of the form $(x, f(x))$, followed by a novel query $x_{\text{query}}$. The goal of the model is to correctly predict the answer as $f(x_{\text{query}})$. Please refer to Appendix G for an example of the full prompt.

We construct $M$ instances of parametric function $f$ for each function class. For each instance, we generate $N$ in-context examples split evenly across all possible labels. We generate the test example randomly and we ensure equal probabilities for its classification into all possible label categories. We do not use any advanced prompting technique when solving ICL tasks. The LLMs only generate answer tokens corresponding directly to classification labels, they do not generate any other tokens during evaluation. We draw all random numbers uniformly from integers between -10 and 10 inclusive. We set $M = 2048, D = 4, N = 32$.

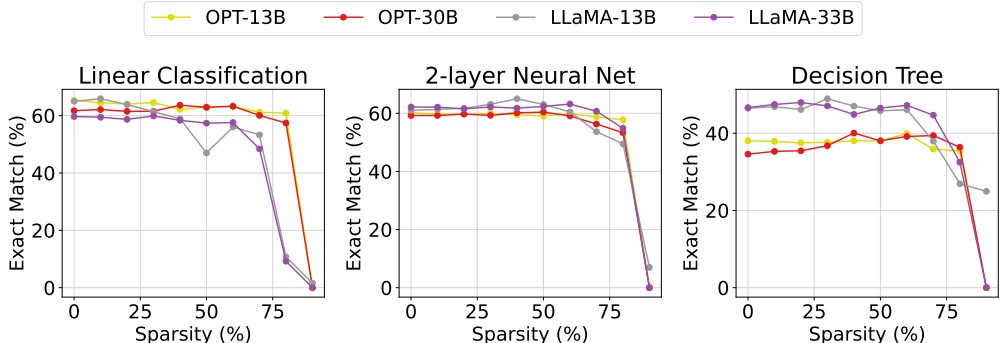

Figure 3: ICL withstands aggresive pruning (60-70% spasity). Accepting a relative average (over 4 models) accuracy drop of 5% w.r.t. the dense models, we can prune to 60%, 60% and 70% on linear classification, 2-layer NN, decision tree tasks, respectively.

**Pruning results.** Figure 19 shows the accuracy of pruned models versus sparsity. We can remove a substantial number of parameters from the models without affecting their ICL capability. Specifically, the average accuracy of the pruned models on linear, 2-layer NN, decision tree classification tasks is within 5% of the dense models up to 60%, 60% and 70% sparsity, respectively.

**Task difficulty.** One trivial explanation for our observation is that perhaps these tasks are so easy that they are solvable even by a severely pruned model. To eliminate this hypothesis, in Appendix C, we increase the input dimension ($D$) of the linear classification task. Even as we increase this to a significant extent (say, to an extent that the task is almost, but not totally, unlearnable), we find that the model maintains its ICL performance even under aggressive pruning.

**Conclusion.** Our results suggest that when pruning LLMs moderately (i.e., to $< 70\%$ sparsity), the model's ICL capability stays largely intact. Thus, unlike fact recall tasks, one can prune models significantly more while maintaining ICL performance.

# 6 DENSE SCALING

For vision models, pruned models behave similarly to smaller dense models trained with similar hyperparameters (Jin et al., 2022). Here, we examine the effects of dense scaling (i.e., choosing from a suite of independently trained models of various sizes) on fact recall vs. ICL capabilities of LLMs.

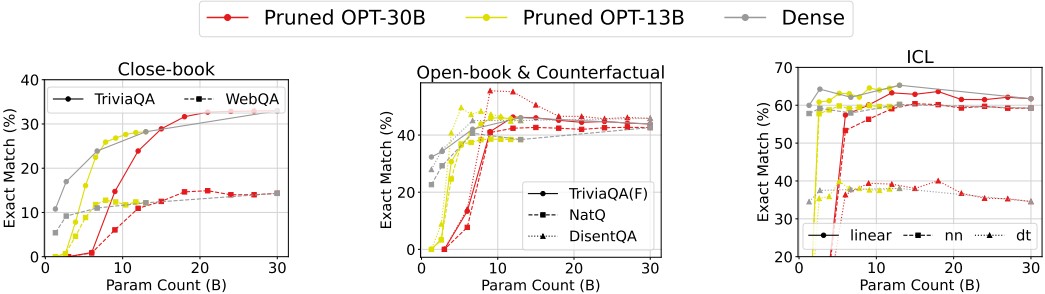

Figure 4: Like pruning, moderate dense down-scaling preserves ICL while harming fact recall. Gray lines show the sequence of dense down-scaled OPT models; we also show the sequence of pruned OPT-30B and OPT-13B models in red and yellow for comparison. Each line style and marker shape combination corresponds to a unique tasks. For the left plot: fact recall is highly sensitive to dense scaling: gray lines shows that, moving from the 30B model to the next largest 13B model leads to more than 5% relative accuracy drop on close-book TriviaQA (solid line/circle marker) and WebQA task (dashed line/square marker), respectively. For the middle plot, open-book/overriding QA accuracy is much less sensitive to dense down-scaling; maintaining the same 5% acceptable accuracy drop, one may down-scale a 30B model to a 6.7B model on TriviaQA(F) (gray, solid line/circle marker), NaturalQuestions dataset (gray, dashed line/square marker) and DisentQA (gray, dotted line/triangle marker). For the right plot, model performance on sophisticated ICL tasks (linear classification, 2-layer NN and decision tree) are highly robust to dense down-scaling.

**Method.** With the same benchmarks as in Sections 4 and 5, we evaluate the accuracy of dense models with different parameter count, but the same pre-training data size. The OPT family of models provides an ideal collection of dense models for our evaluation: Zhang et al. (2022) trained all OPT models with the same number of tokens (300b).

**Results.** Figure 4 shows that fact recall capability is highly sensitive to dense scaling – e.g., focusing on scaling downwards, moving from the 30B model to the next largest 13B model leads to more than 5% relative task accuracy degradation on close-book TriviaQA and WebQA task. However, open-book/overriding QA accuracy is much less sensitive to dense scaling. Specifically, maintaining the same 5% acceptable relative accuracy degradation with respect to the 30B models, one may replace a 30B model on TriviaQA(F), NaturalQuestions and DisentQA dataset with a 6.7B model. Figure 4 also shows that sophisticated ICL capability is similarly robust to dense scaling. Specifically, the accuracy difference between the largest (30B) and the smallest (1.3B) dense OPT model we consider is 1.8%, 1.4% and 0.1% on linear, 2-layer NN and decision tree classification tasks, respectively.

**Conclusion.** Like pruning, changing a dense model's size more readily affects its capability to retrieve facts from pre-training than to process information from context. We hypothesize that this effect stems from scaling in general – be it pruning or using a dense model of different size.

# 7 CLOSING DISCUSSION

We study the effects of scaling model size via pruning and dense scaling on two core capabilities of LLMs that underpin their practical success: the capability to recall facts and the capability to perform in-context learning (ICL). In both cases, we find disparate effects on the two capabilities. Moderate pruning ($> 30\%$ sparsity) harms fact recall, and yet the capability to learn from a few input-output examples from context withstands aggressive pruning (up to 60–70% sparsity). The same disparity arises when changing the width and depth of dense (independently-trained) models.

What could explain this disparity? We conjecture that the number of parameters required to store a set of facts must scale in proportion to the number of independent facts. On the other hand, a smaller set of parameters acting as a universal gradient descent module (Von Oswald et al., 2023) may accomplish many kinds of in-context learning tasks. Verifying these hypotheses theoretically is an important direction for future work. Our findings also invite various research directions:

**Pruning & interpretability.** Our findings suggest, remarkably, a relatively small fraction of weights are responsible for ICL performance. This observation could prove to be valuable for enhancing the interpretability of LLMs, reviving a decades-old motivation behind work in pruning (Mozer & Smolensky, 1988). In particular, pruning may help better localize the weights responsible for ICL and fact recall capabilities, and complement recent approaches such as identifying circuits consisting of attention heads responsible for specific capabilities (Wang et al., 2023), and explaining neuron functionalities using large language models themselves (Bills et al., 2023). In Appendix F, we present preliminary results comparing the importance of feedforward (FFW) versus attention layers for maintaining ICL and fact recall capabilities. Our evaluation shows that while FFW and attention layers show similar importance for ICL, FFWs layers show greater importance for fact recall.

**Memory augmentation.** Our observations advocate for *memory augmentation* as a promising way to improve the trade-off between computational cost and task accuracy. Memory augmentation techniques present helpful facts to the model by augmenting them directly in the context. Thus, rather than having the LLM to rely on fact recall from pre-training — an ability that is degraded under downscaling — we can delegate fact-retrieval to a separate retrieval model (Borgeaud et al., 2022; Guu et al., 2020), and allow the LLM to focus on retrieving the fact from the context—which is an ability that *is* preserved under down-scaling.

**Limitations.** We validated claims on a large number of benchmarks (a total of 8 tasks and 6 models). However, since our work is empirical in nature, our observations may not generalize to all tasks and LLMs. Furthermore, we focused only on pruning algorithms that scale to the size of contemporary LLMs, as we believe that scalability is essential for the wide adoption of any pruning algorithm. Even though more sophisticated pruning algorithms exist (Renda et al., 2020), they typically require re-training — re-running the training of these large-scale language models for every sparsity level we examine. The cost of such an experiment is therefore beyond our means.

**Acknowledgements.** We thank Zack Ankner, William Brandon, Ellie Y. Cheng, Jesse Michelle, Nicole Mitchell, Aniruddha 'Ani' Nrusimha, Alex Renda, Daniel M. Roy, Logan Weber for their helpful feedback on this work. This work was supported in-part by the Sloan Foundation, Google Research, the MIT-IBM Watson AI Lab, Apple, and SRC JUMP 2.0 (CoCoSys).

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

| Name/Sparsity | 0.0 | 10.0 | 20.0 | 30.0 | 40.0 | 50.0 | 60.0 | 70.0 | 80.0 | 90.0 |
|---|---|---|---|---|---|---|---|---|---|---|
| OPT-13B | 11.5 | 11.5 | 11.6 | 11.6 | 11.9 | 12.4 | 13.7 | 17.8 | 45.3 | 40362.3 |
| OPT-30B | 10.9 | 10.9 | 10.9 | 11.0 | 11.1 | 11.4 | 12.2 | 14.6 | 35.4 | 4187.5 |
| LLaMa-13B | 6.6 | 6.6 | 6.7 | 6.8 | 7.1 | 7.8 | 10.0 | 20.0 | 59.9 | 822.1 |
| LLaMa-33B | 5.9 | 6.0 | 6.0 | 6.1 | 6.4 | 7.0 | 8.5 | 13.7 | 36.1 | 355.9 |

Table 2: Pruning's effect on perplexity.

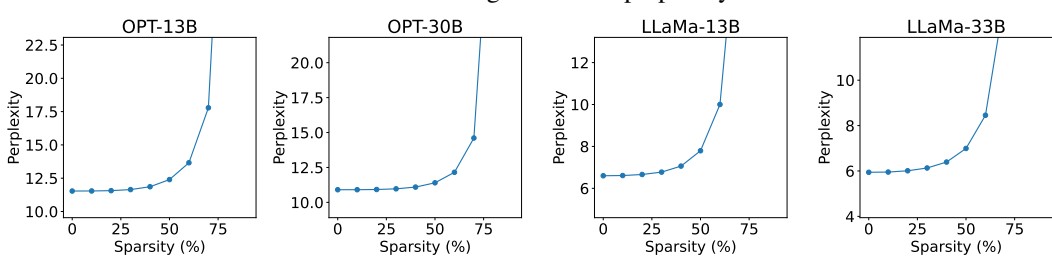

Figure 5: Pruning's effect on model perplexity.

## A    PRUNING PERPLEXITY EVAL

In Table 2 we present the full range of pruning's effect on the perplexity of models we consider in this work. We obtain perplexity results by running the pruned model on a randomly sampled subset of C4 validation set, following the precedent of Frantar et al. (2023).

## B    CONSTRUCTING PARAMETERIZED FUNCTIONS IN ICL EVALUATION

While it may be unreasonable to expect the model to infer any arbitrary $f$, we focus on three natural classes of functions from which we pick $f$: the class of linear, 2-layered neural network, and decision tree models. These classes are defined as follows: (a) *Linear.* For each task instance, we generate a distinct random hyperplane in the $D$-dimensional input space as the decision boundary. We label each example as positive/negative depending on the side of the decision boundary it falls on. (b) *2-layer NN.* For each task instance, we generate a 2-layer neural network mapping a $D$-dimensional input vector $x$ to a binary label $y$ with the following form: $W_2\sigma(W_1x)$, where $W_1$, $W_2$ are $D \times D$, $D \times 2$ matrices correspondingly. We draw elements within these two matrices from independent, Gaussian distributions with zero-mean and unit variance. We use ReLU as the activation function $\sigma$. (c) *Decision tree.* For each task instance, we construct a full depth-2 binary decision tree, with each of its four leaf nodes representing one possible label. This maps $D$-dimensional input vectors to the said labels. We assign each non-leaf node an index from 0 to $D - 1$. We evaluate the tree by traversing from the root, comparing the input vector's value at each node's associated index to determine the next step: if negative, we go left; if zero or positive, we go right. Reaching a leaf node marks the end of traversal, and the corresponding label represents the evaluation result.

## C    INCREASING INPUT DIMENSION OF ICL EVALUATION

In this section, we further evaluate the effect of pruning on ICL tasks with different input dimensions.

**Method.**    In Section 5, we evaluate pruning's effect on learning linear classification tasks in-context. The task is to classify 4-dimensional input vectors. Here, we increase the input vector dimensions to 8 and 16, and observe whether our observation about pruning's effect generalize to larger input dimenions.

**Results.**    Figure 6 presents the relationship between task accuracy and sparsity for linear classification task. We increase the input dimension from 4 to 8 and 16. We observe that tolerating the same 5% relative drop in the average accuracy among all four models with respect to the dense model, we can prune to 60%, 80% sparsity on linear classification task with 4, 8 dimensional inputs respectively.

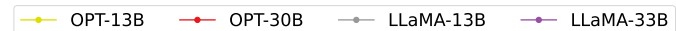

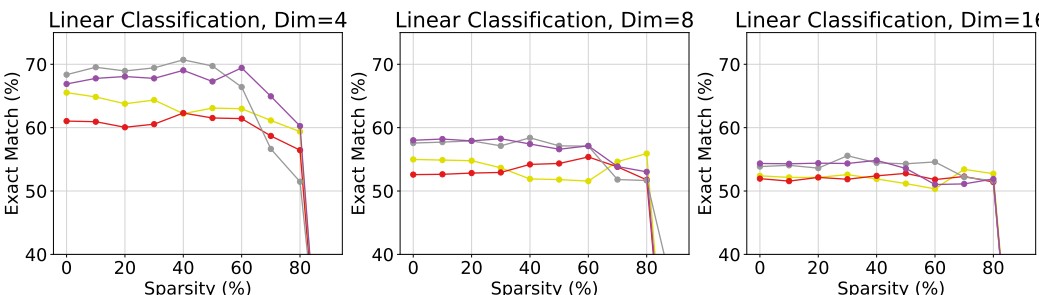

Figure 6: Pruning's effect on ICL linear classification tasks with input dimension of 4, 8 and 16. We observe that even with increased input dimension, ICL ability of LLMs continue to be robust to aggresive pruning.

Dense models perform close to chance on 16-dimensional inputs, thus we refrain from drawing conclusions based on these results.

**Conclusion.** Even with higher dimensional inputs, model's ICL ability remains resilient to aggressive pruning ($>= 60\%$ sparsity).

## D  ADDITIONAL PRUNING ALGORITHM EVALUATION

In our main paper, we focus on a single pruning algorithm SparseGPT (Frantar & Alistarh, 2023). In this section, we study whether our observation that scaling down LLM size affects fact recall more readily than ICL generalize to another pruning algorithm.

**Method.** We repeat key experiments with another pruning algorithm called Wanda (Sun et al., 2024). Notably, unlike SparseGPT, Wanda does not update the remaining weights after weights removal. The author of Wanda shows that at 50% sparsity, Wanda achieves accuracy that is competitive with SparseGPT.

**Results.** We observe the same disparate effects from pruning on fact recall versus ICL: while moderate pruning hurts fact recall, ICL survives to higher sparsity. Specifically, accepting the same 5% relative drop in accuracy as we did for SparseGPT results, one may remove 30%, 40% and 50% weights on TriviaQA(Closebook), TriviaQA(Openbook) and Linear Classification tasks. Unsurprisingly, given that Wanda is computationally less expensive, it underperforms SparseGPT at high sparsities on ICL tasks.

**Conclusion.** With our experiments repeated with another pruning algorithm, we show that our observation is general.

## E  PYTHIA MODELS EVALUATION

In this section, we study whether our observation that scaling down LLM size affects fact recall more readily than ICL generalize to another model family: Pythia (Biderman et al., 2023b).

**Method.** We repeat key experiments with Pythia model family. With SparseGPT, we pruned two variants of the Pythia-12B model, the original and a "deduped" variant where the model is trained on a deduplicated dataset. They're the most capable models in the Pythia family.

**Results.** Figure 8 shows consistent disparate effect of pruning on fact recall versus ICL. Specifically, accepting the same 5% accuracy drop as in all our results, one can remove 20%, 30% and 50% weights for TriviaQA(Closebook), TriviaQA(Openbook) and in-context linear classification tasks.

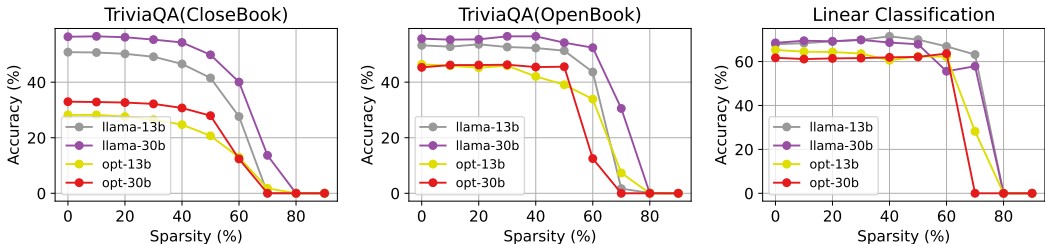

Figure 7: An alternative pruning algorithm (Wanda) shows the same patterns of accuracy drop as findings in our original paper: moderate pruning hurts fact recall (e.g., left, TriviaQA in Closebook setting) while ICL (e.g., right, in-context linear classification) survives to higher sparsity. Specifically, accepting a 5% relative drop in accuracy, one may remove 30%, 40% and 50% weights on TriviaQA(Closebook), TriviaQA(Openbook) and Linear Classification tasks, respectively.

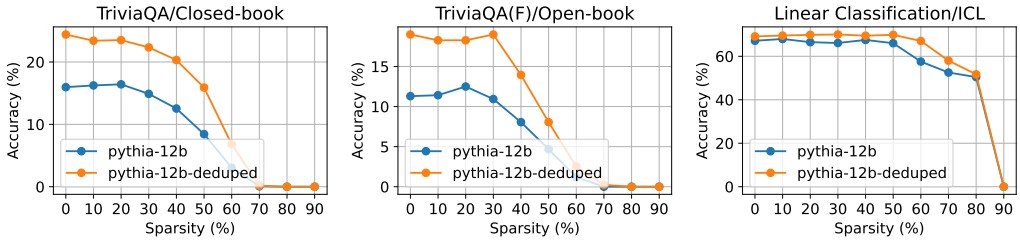

Figure 8: Two variants of Pythia-12B model shows the same patterns of accuracy drop as findings in our original paper: moderate pruning hurts fact recall (e.g., left, TriviaQA in Closebook setting) while ICL (e.g., right, in-context linear classification) survives to higher sparsity. Specifically, accepting a 5% relative drop in average accuracy, one may remove 20%, 30% and 50% weights on TriviaQA(Closebook), TriviaQA(Openbook) and Linear Classification tasks, respectively.

**Conclusion.** Our observation is robust to the choice of model families.

> Answer these questions:
> Q: In Scotland a bothy/bothie is a?
> A: House
> Q: Which 90s sci fi series with James Belushi was based on Bruce Wagner's comic strip of the same name?
> A:

Figure 10: Examples Prompt for Question-answering Task without Context.

## F    PRUNING ATTENTION AND FEED FORWARD LAYERS

A growing line of work (Dai et al., 2021; Meng et al., 2022) suggests that feed forward (FFW) layers and the attention layers are responsibly for distinct capabilities considered in this paper — namely, fact recall and in-context learning. To test this, we exclusively prune either type of layers and examine how the model's capabilities deteriorate on fact recall and ICL tasks.

**Method.** We prune a LLaMA-13B model. We either exclusively prune attention layers or exclusively prune FFW layers and observe their effects on accuracy of TriviaQA task (fact recall) and linear classification task (ICL). We plot the accuracy as a function of *module sparsity*. Module sparsity refers to the fraction of parameters pruned from a specific type of layers (either attention or FFW), with respect to the total number of parameters of that type.

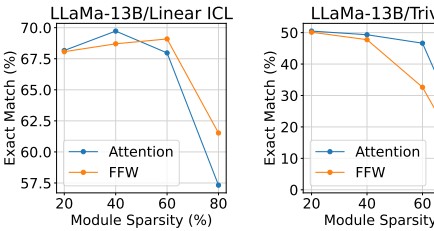

**Results.** Figure 9 shows that while attention and FFW layers appear equally important for ICL, FFW layers appear more important for fact recall. For example, pruning 60% FFW layers lead to 14% more accuracy degradation than pruning 60% of attention layers.

**Conclusion.** Attention and FFW layers show similar importance for ICL; and FFW layers show greater importance for fact recall.

(a) Linear Classification    (b) TriviaQA Close-book

Figure 9: Effects of pruning only attention layers versus pruning only FFW layers on fact recall (TriviaQA) versus ICL (Linear Classification). Module sparsity computes the fraction of parameters of the specified module (attention or FFW) that pruning removes. Attention and FFW layers appear equally important for ICL. FFW layers appear more important than attention layers for fact recall.

## G    EXAMPLE PROMPTS

In this section, we present example prompts for Q&A tasks in Section 4 and ICL tasks in Section 5.

### G.1    QUESTION-ANSWERING WITHOUT CONTEXT.

Our prompt follows the precedent of Touvron et al. (2023). We show our prompt for Q/A tasks without context in Figure 10.

### G.2    QUESTION-ANSWERING WITH CONTEXT.

We show an example prompt we use to evaluate Q/A task with the necessary information to answer the question available in-context in Figure 11.

### G.3    ICL TASK.

We present the prompt we use to evaluate ICL on linear classification task in Figure 12. Evaluations for other ICL tasks uses analogous prompts.

Answer these questions:
Context: Scattered across the Highlands and rural areas of Scotland, a bothy is a small house that can be used by anyone.
Q: In Scotland a bothy/bothie is a?
A: House
Context: "Roses Are Red (My Love)" is a popular song composed by Al Byron and Paul Evans. It was recorded by Bobby Vinton and was his first hit.
The song was released in April 1962. It reached No. 1 in Australia, New Zealand, Norway, the Philippines, South Africa, and the United States, and was a major hit in many other countries as well. The song topped the Billboard Hot 100 singles chart on July 14, 1962, and remained there for four weeks. The single was also the first number-one hit for Epic Records. Billboard ranked the record as the No. 4 song of 1962.
Vinton found the song in a reject pile at Epic Records. He first recorded it as an R&B number, but was allowed to re-record it in a slower more dramatic arrangement, with strings and a vocal choir added.
Ronnie Carroll version
In the UK, a cover version by Ronnie Carroll reached No. 3 on the Record Retailer chart on August 8, 1962, the same week that the Bobby Vinton record peaked at No. 15. It peaked at No. 7 in the very first Irish Singles Chart published in October 1962.
Other versions
The song was recorded by Jim Reeves in 1963 and released on the album Gentleman Jim, one of the last albums released while he was still alive. While it did not chart in the US, it became a minor hit in Norway and Germany.
The song was covered by Singaporean female artist Zhuang Xue Fang, in edited Standard Chinese lyrics written by Suyin under title name of, with Ruby Records in 1967.
In 1962, an answer song, entitled "Long As The Rose Is Red", was recorded by Florraine Darlin. The song spent seven weeks on the Billboard Hot 100, reaching No. 62, while reaching No. 15 on Billboards Easy Listening chart. It was released by Epic Records (single #9529) and was also produced by Robert Morgan.
Charts
Bobby Vinton version
Ronnie Carroll version
Q: Which singer had a big 60s No 1 with Roses Are Red?
A:

Figure 11: Examples Prompt for Question-answering Task with Context.

```
[5, -5, -1, 6] = -1
[3, 5, -9, -2] = 1
[-10, 5, -9, -10] = 1
[7, 0, -4, -7] = -1
[-2, -2, 9, -3] = -1
[9, -5, -5, -1] = -1
[-4, -1, -10, -8] = 1
[-10, -1, -3, 6] = -1
[3, 6, -5, -8] = 1
[-1, 7, -2, 6] = -1
[-5, 8, -1, -9] = 1
[-6, -1, -6, -9] = 1
[1, 0, -10, -9] = 1
[-5, 5, -3, 6] = -1
[-4, 6, -10, -7] = 1
[-4, 5, -6, -10] = 1
[-10, 5, -3, -9] = 1
[-7, -2, -7, -9] = 1
[-7, 1, -7, 4] = -1
[-2, 2, -3, -10] = 1
[1, -4, -4, 9] = -1
[7, 0, 6, -10] = -1
[2, 7, -6, -1] = 1
[1, -2, -10, -9] = 1
[-8, 3, 7, -7] = -1
[0, 7, -3, -8] = 1
[0, 2, 0, 0] = -1
[-7, 9, 2, 8] = -1
[1, -8, -9, 6] = -1
[-5, 1, -6, -10] = 1
[8, 8, 1, -4] = -1
[-3, 6, 2, 2] = -1
[-3, 9, 0, -5] =
```

Figure 12: Examples Prompt for ICL on Linear Classification Task.

## H   ANALYSIS ON ACCURACY IMPROVEMENT

On many tasks, such as ICL on algorithmic tasks in Section 5, we observe accuracy improvement from pruning. In this section, we provide further analysis on this phenomenon.

**Method.**   We focus on the ICL task of check-sorting and check-even from Section 5, where the pruned LLaMA models see significant accuracy improvement. We analyze the prediction of the pruned model, in relation to the following two answers: 1). *Answer without context.* We compute the classification output with the highest log likelihood, without all the in-context examples. The prompt thus looks like "[1, 0, 3, -5]=" without any other information such as in-context training examples. The LLM thus classifies this input array exclusively based on likelihood of a classification label as a continuation of the prompt. We denote this classification outcome as the answer without context. 2). *Ground-truth answer.*

**Results.**   In Figure 13, we present the percentage of time when the prediction of the pruned model matches the answer without context versus the ground truth. We observe that accuracy improvement often mirrors a departure of the model's prediction from answer without context (the solid line dropping whilst the dashed line raising). This suggest that for some pruned models (for example LLaMA-30B at 60% sparsity), the presence of in-context examples changes the prediction of the LLM more than the dense models.

**Conclusion.**   We hypothesize, without testing, that pruning may improve task accuracy by enhancing the effect of contextual information on its prediction.

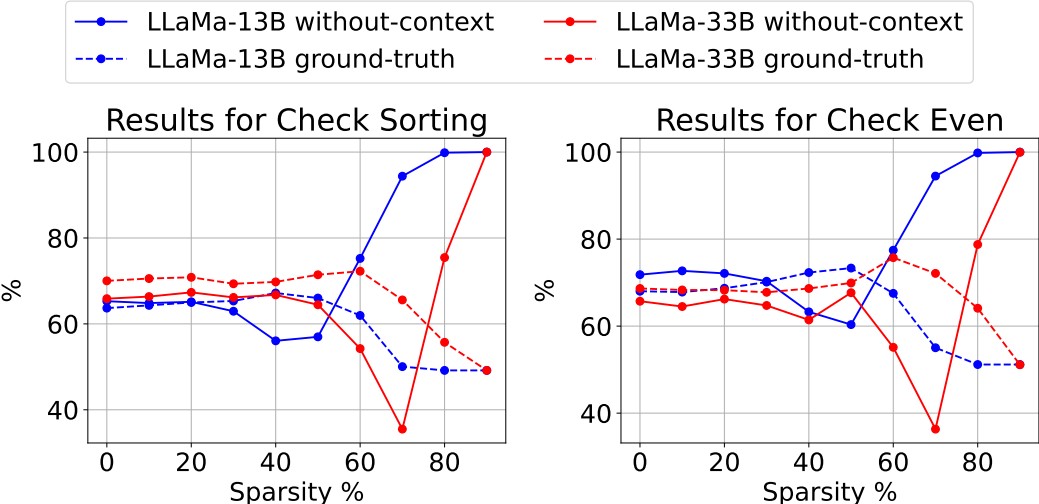

Figure 13: Analyzing the accuracy improvement of pruned model on ICL tasks.

# I   CHARACTERIZING ERRORS FROM NUMERICAL PRECISION

All our experiments run on TPU v3. The TPU hardware does not support float16 (IEEE 754 half precision floating point format), which are the formats used for training the OPT/LLaMA models. We thus use a closely related floating point format called bfloat16 (Brain Float) instead to evaluate these models. Both bfloat 16 and float16 are floating point numbers requiring 16 bits to represent, the key difference is that bfloat16 format can represent numbers in a slightly wider range of values, but with slightly reduced precision compared with float16. For this reason, we expect our results to contain small systematic errors. In this section, we characterize the difference between our results in bfloat16 and standard results in float16.

**Perplexity and accuracy results.**   In Figure 14, we plot the perplexity and accuracy in next token prediction in bfloat16 on TPU v3 and float16 on A100. We observe that the average accuracy difference is 1.0%, 0.89%, 0.70% and 0.68% for OPT-13B, OPT-30B, LLaMA-13B and LLaMA-33B. The perplexity grows to very large numbers at high sparsity, so we report the average perplexity difference for sparsity level less than or equal to 50%. The average perplexity difference is: 0.86, 0.69, 0.19, 0.13. We note that 1). the difference in accuracy and perplexity due to numerical precision is systematic. The accuracy of next token prediction raises by a similar amount across all sparsity levels. 2). the difference in next token prediction accuracy is small (<1%).

**OPT.**   OPT models see higher difference in accuracy and perplexity. This is because its GPU implementation uses mixed precision arithmetic, where the attention score computes in float32. Our compiler and runtime, however, does not support mixed precision arithmetic, we therefore compute entirely within bfloat16.

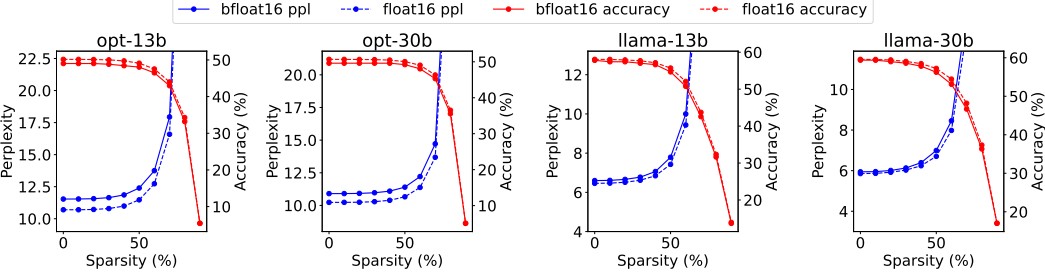

Figure 14: Characterizing the difference between float16 and bfloat16.

# J   BEAM SEARCH + TOLERANT ANSWER MATCHING

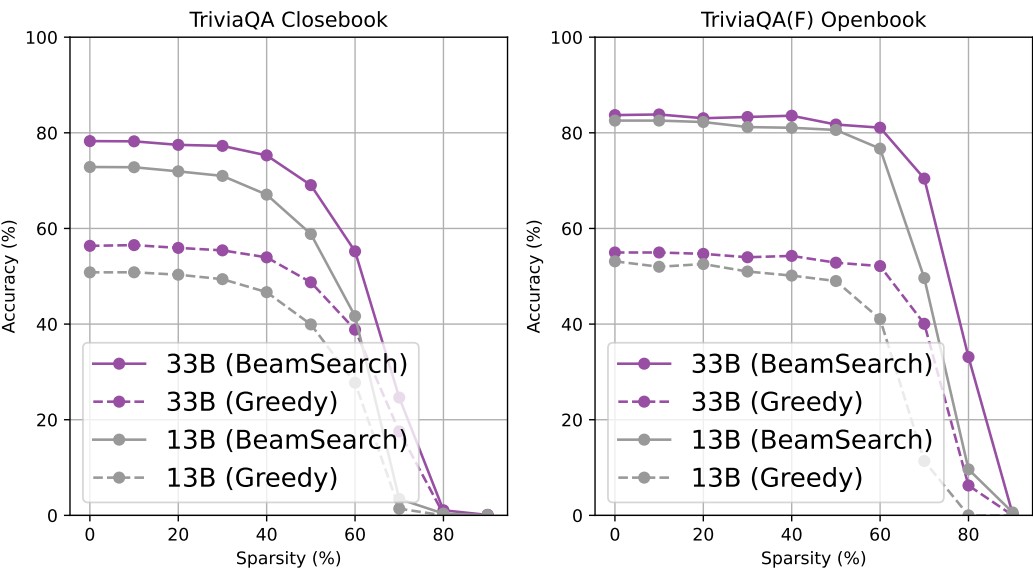

Figure 15: Reproducing TriviaQA Closebook and TriviaQA(F) Openbook results using beam search (beam size=3) and more tolerant answer matching technique on LLaMA-13B/33B: we generate 32 tokens and check if the correct answer appear anywhere in the generated response. Our conclusion remains unaffected: using our current experimental design, averaging over two LLaMA models, one can prune 30% and 40% weights on Closebook and Openbook TriviaQA task, respectively. Using beam search, one can prune 30% and 50% weights on Closebook and Openbook TriviaQA tasks, respectively. The ability to retrieve answers from provided context remains more resilient to pruning than the ability to recall information learnt during pre-training.

## K  MORE TASKS

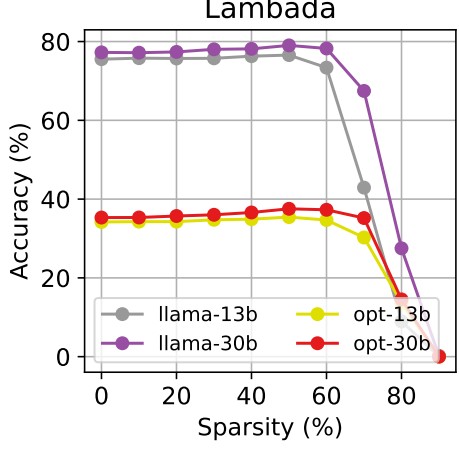
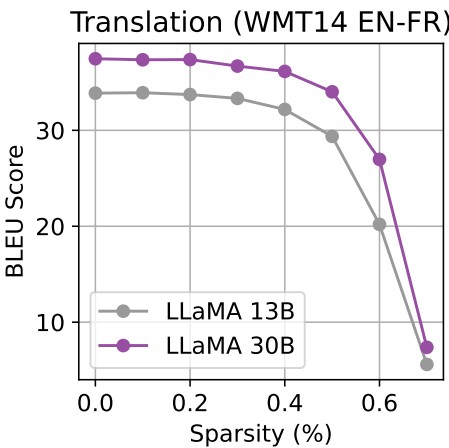

Figure 16: Lambada    Figure 17: Translation

## L  IN-CONTEXT LEARNING ON ALGORITHMIC TASKS

**Method.**  We use the following ICL task and for each task, generate a total of N task instances. Each task instance consists of a set of in-context examples along with a test input. a). *Check Sorting.* The task is to determine, whether a D-element input array is sorted. We label each example as positive/negative depending on whether the input array is sorted or not.b). *Check Contains Even Number.* The task is to determine, whether a D-element input array contains an even number. We label each example as positive/negative depending on whether the input array contains an even number. We present the pseudo code solution to these algorithmic tasks in Figure 18.

All task instances consist of K positive in-context examples and K negative in-context examples. We draw all random numbers from integers between -10 to 10 inclusive. We set N=2048, D=4, K=16.

**Sparsity results.**  Figure 19 shows the accuracy of pruned models versus sparsity. We can remove substantial number of parameters from the models without affecting their ICL ability. Specifically, the average accuracy of the pruned models on check sorting and check contains even number is within 5% of the dense models up to 60% and 70% sparsity respectively.

**Conclusion.**  Our results suggest that when pruning to moderate sparsity levels (60-70%), the model's ICL ability stays largely intact. Compared with pruning on fact recall tasks, one can prune to much higher sparsity levels while maintaining or even improving ICL performance.

```
1  def is_sorted4d(A):
2    s = True
3    s = s and A[0] <= A[1]
4    s = s and A[1] <= A[2]
5    s = s and A[2] <= A[3]
6    return s
7
```

```
1  def contains_even4d(A):
2    c = False
3    c = c or A[0] % 2 == 0
4    c = c or A[1] % 2 == 0
5    c = c or A[2] % 2 == 0
6    c = c or A[3] % 2 == 0
7    return c
8
```

Figure 18: Algorithmic tasks for in-context learning evaluation. (*Left*) Check if a 4-D vector is sorted. (*Right*) Determine whether a 4-D input vector contains an even number.

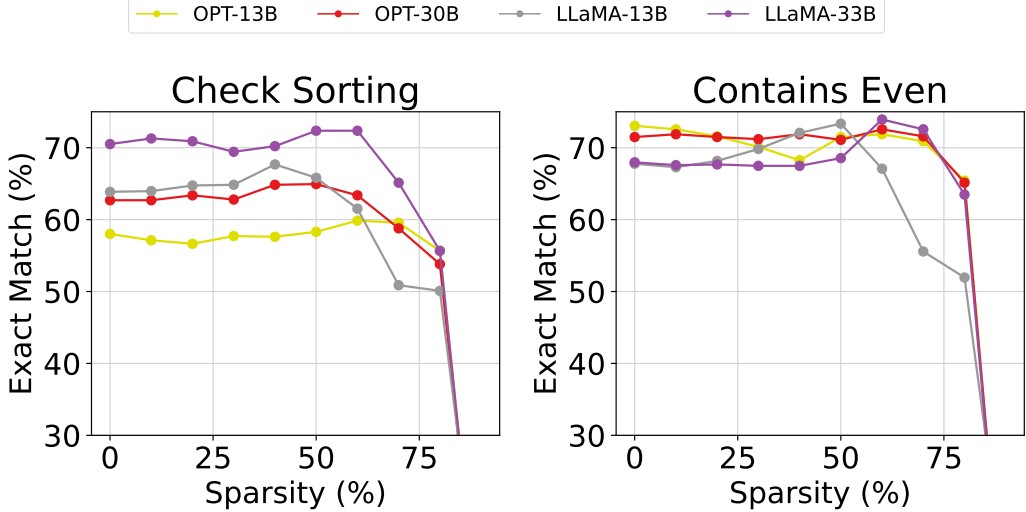

Figure 19: ICL withstands aggresive pruning (60-70% spasity). Accepting a relative average (over 4 models) accuracy drop of 5% w.r.t. the dense models, we can prune to 60% and 70% on check sorting and check contains even number tasks, respectively.

## M    PRUNING ALGORITHMS DESCRIPTION

In this section, we present implementation details of pruning algorithms we evaluate in this work.

### M.1    SPARSEGPT.

The SparseGPT algorithm prunes weights in a language model layer by layer. For each layer's weight $W_l$, where $l$ is the layer index, it produces a binary mask $M_l$ and a updated set of weights $\hat{W}_l$ to minimize the layerwise reconstruction loss $\|W_l X - (M \odot \hat{W}_l) X\|_2^2$, where $\odot$ denotes element-wise multiplication, and X correspond to inputs to layer $l$. Such inputs are computed using a small set of calibration data, randomly drawn from the training dataset. Note that rows in $W_l$ do not interact with one another, SparseGPT thus prunes each row independently.

**Weight selection and repair.**    SparseGPT builds on optimal brain surgeon (OBS) (Hassibi et al., 1993; Frantar et al., 2021), a classical approach in weight pruning. For a given weight row $w$, upon removing an element at position $m$, OBS suggests the optimal weight update $\delta_m$ to compensate for the effects of removed weights, and characterize the associated reconstruction error $\epsilon_m$ as follows:

$$\delta_m = -\frac{w_m}{[H^{-1}]_{mm}} \cdot H^{-1}_{:,m}, \quad \epsilon_m = \frac{w_m^2}{[H^{-1}]_{mm}}. \tag{1}$$

The SparseGPT thus prunes rows of $W_l$ independently and in parallel. Within each row, the algorithm prunes weights in vectors of size $B = 128$. The algorithm selects weights for removal based on the lowest reconstruction error $\epsilon$ After removing weights, SparseGPT iteratively applies the OBS weight update. An important difference between SparseGPT and OBS is that while OBS adjusts all remaining weights to compensate for removed weights, SparseGPT only updates a subset of remaining weights to reduce the computational complexity associated with computing the Hessian inverse. This complexity reduction enables SparseGPT to scale to very large language models (175B). See pseudocode below for details about the exact subset of remaining weights updated by SparseGPT.

**Pseudocode**    The following pseudocode is a simplified version of SparseGPT largely based on Algorithm. 1 of Frantar & Alistarh (2023). This algorithm prunes the layer weight matrix $W$ to $p\%$ unstructured sparsity given inverse Hessian $H^{-1} = (XX^T)^{-1}$, weight vector size $B$.

---

**Algorithm 1** The SparseGPT algorithm.

1:  $M \leftarrow 1_{d_{row} \times d_{col}}$           ▷ binary pruning mask
2:  $E \leftarrow 0_{d_{row} \times B}$          ▷ block pruning errors
3:  **for** $i = 0, B, 2B, \ldots$ **do**
4:      $M_{:,j:(j+B_s)} \leftarrow$ mask of $(1 - p\%)$ weights $w_c$ in $W_{:,j:(j+B_s)}$ with largest $w_c^2/[H^{-1}]_{cc}$
5:      # Apply the OBS update to a subset of remaining weights.
6:      **for** $j = i, \ldots, i + B - 1$ **do**
7:          $E_{:,j-i} \leftarrow W_{:,j}/[H^{-1}]_{jj}$
8:          $E_{:,j-i} \leftarrow (1 - M_{:,j}) \cdot E_{:,j-i}$
9:          $W_{:,j:(i+B)} \leftarrow W_{:,j:(i+B)} - E_{:,j-i} \cdot [H^{-1}]_{j:(i+B)}$
10:      **end for**
11:      $W_{:,i:(i+B)} \leftarrow W_{:,i:(i+B)} - E \cdot H^{-1}_{i:(i+B),(i+B)}$
12: **end for**
13:  $W \leftarrow W \cdot M$          ▷ set pruned weights to 0

---

## M.2   WANDA

Wanda(Sun et al., 2024) similarly prunes a language model layer by layer. Within each layer, it independently prunes individual rows in the weight matrix. The inputs to a layer, denoted as $X$, have the shape $(N, L, C_{in})$. Here, $N$, $L$, and $C_{in}$ refer to the batch size, sequence length, and input hidden dimension size, respectively. The inputs to the layer $W$ derive from a small set of calibration data. The weight matrix to prune has shape $(C_{in}, C_{out})$. $C_{in}$ and $C_{out}$ refer to the input and output hidden dimension sizes. Wanda scores each weight $W_{i,j}$ using the following equation:

$$S_{ij} = |W_{ij}| \cdot \|X_j\|_2 \, ,$$

where $|.|$ denotes the absolute value function. Wanda removes $p\%$ of the weights with the lowest scores in each row $W_i$. Notably, unlike SparseGPT, there is no weight update after weights removal.

## N   STRUCTURED PRUNING (LLM-PRUNER) RESULTS

In this section, we test our findings using structured pruning algorithm LLM-Pruner (Ma et al., 2023).

**Configuration.**   We use LLaMA7B/13B instead of 13B/30B in the main paper because LLM-Pruner appears to consume significantly more GPU memory than SparseGPT, and 30B model does not fit within our hardware setup.

Following the recommended setup of Ma et al. (2023) in their code repository, we skip the first 4 layers and last 2 layers for all LLaMA models we test. Following Frantar & Alistarh (2023), we sample calibration data of maximum sequence length (2048 tokens) from C4 dataset. The number of samples in the calibration data is configured to maximum allowed by our system memory size (167 GB CPU memory + 80GB GPU memory), which is 2 for 7B and 1 for 13B LLaMA model.

Since Ma et al. (2023) focuses on the low-sparsity regime (0-20%) in the paper, we similarly focus on this regime of sparsity. Specifically, we prune each of the prunable layers (which excludes 4 initial and 2 final layers) to 5%, 10%, 15%, 20%, and 25% sparsity. We refer to this sparsity ratio as local sparsity, since it refers to the sparsity ratio within prunable layers only. In Figure 20, we report global sparsity to remain consistent with our main paper – global sparsity refers to the number of remaining parameters divided by the total number of parameters of the full model.

**Results.**   Figure 20 shows that our main conclusion that fact recall deteriorates faster than ICL under down-scaling remains valid with the structured pruning algorithm. Specifically, averaging over two models, we can prune to 0%, 5% and 20% local sparsity on Closebook (TriviaQA), Openbook (TriviaQA(F)) and ICL linear classification tasks, respectively, without dropping more than 5% relative accuracy.

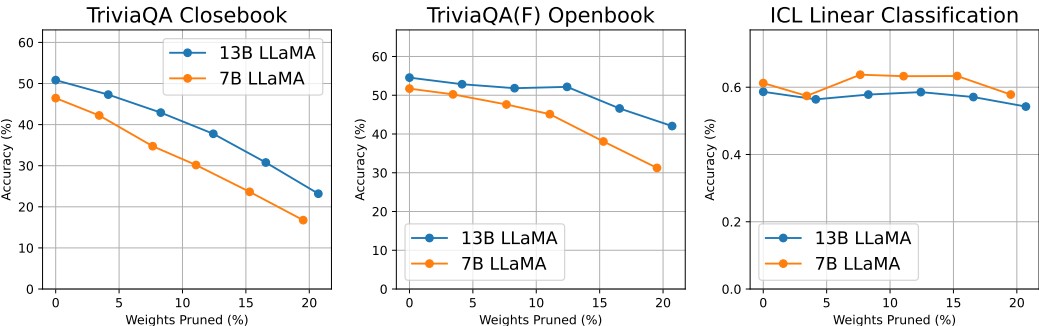

Figure 20: Structured pruning results.

