# OpenReview forum: "The Cost of Scaling Down Large Language Models: Reducing Model Size Affects Memory before In-context Learning"
_ICLR.cc/2024/Conference — ICLR 2024 poster_

### Official Review · Reviewer_B31e · 2023-10-29

**Soundness:** 4 excellent
**Presentation:** 3 good
**Contribution:** 4 excellent
**Rating:** 6
**Confidence:** 4

**Summary:**

The paper investigates the effects of pruning and down-scaling large language models (LLMs) on the model capabilities. Specifically, the authors focus on two abilities of modern LLMs: (1) the ability to process information stored in the weights (fact recall) and (2) the ability to process information that is available in context. To evaluate down-scaled models on these capabilities, they use a suite of benchmarks covering four tasks, open-book QA, closed-book QA, overriding QA, and learning tasks (i.e., model needs to understand underlying function based on examples given as in-context learning). Experiments on 6 base LLMs, each with 9 different sparsity levels demonstrate different model behavior in terms of the two capabilities. Model ability to process information in weights degrades with moderate level of pruning (>30%), while model ability to process information in context does not really degrade even with aggressive pruning (up to 70%).

**Strengths:**

Overall, I think the paper discusses an important question regarding the trade-offs of having smaller-scaled models and its impact to model capabilities. The experiments are well-thought, with the use of different benchmark tasks to isolate different model capabilities being tested and the use of different base LLMs to see that the effects are similar across different model families. I think the main findings of this paper will be useful for future work in this area. The paper is well-written.

**Weaknesses:**

- Although down-scaling and pruning are the main topic of the paper, the technical details on methods used is very limited (even in the Appendix too). If space is an issue, I would suggest to cut down the paper motivation which is repeated multiple times throughout the paper.
- Relatedly, there is very little discussion regarding down-scaling vs. pruning. For general readers it would be helpful to understand what are the difference between the two, and is one a specific version of the other?

**Questions:**

- For learning tasks evaluation, why only consider task with scalar values as labels? I understand this needs to be something that model can generalize through the examples, but if we focus on language capability of the model, I would expect that a natural language task is used instead.
- For ICL results, it seems the performance drops significantly (not gradually) from 70% above, do you have intuition why?
- As the paper only use some particular pruning methods, do you have any opinion on whether the same findings will hold for other pruning methods? It would be good to have a short discussion on the difference between them.

**Things to improve the paper**
- Section 5 is hard to follow without examples, there are multiple notations without explanation, e.g. K (page 7), x, D=4, N=32, etc
- Would be good to release the version of datasets that are used for benchmarking

---

> ### Author Response · Authors · 2023-11-17
> **Response**
>
> We thank the reviewer for the insightful feedback and questions!
>
> > Although down-scaling and pruning are the main topic of the paper, the technical details on methods used is very limited (even in the Appendix too). If space is an issue, I would suggest to cut down the paper motivation which is repeated multiple times throughout the paper.
>
> We are working on including more details about the pruning techniques to make our work self-contained. Will update once we apply the edits.
>
> > Relatedly, there is very little discussion regarding down-scaling vs. pruning. For general readers it would be helpful to understand what are the difference between the two, and is one a specific version of the other?
>
> We can view dense down-scaling as a form of structured pruning. While SparseGPT prunes at the granularity of individual weight, we can view dense-downscaling as pruning at the granularity of entire neurons and entire layers. Some important differences between dense-scaling and pruning include:
> - The order of weights removal and training – viewing dense downscaling as structured pruning, weights removal happens before training. However, for pruning, weight removal happens after training (and is followed by repair).
> - Granularity of weights removal – dense downscaling removes entire neurons & layers, whereas pruning removes individual weight.
>
> > For learning tasks evaluation, why only consider task with scalar values as labels? I understand this needs to be something that model can generalize through the examples, but if we focus on language capability of the model, I would expect that a natural language task is used instead.
>
> Please refer to our global response, additional task evaluation section. We provide additional results on the LAMBADA benchmark, a natural language based ICL task.
> Our focus on synthetic ICL tasks (thus scalar/synthetic labels) is driven by our goal to disentangle fact recall from ICL tasks. Many existing natural language based ICL tasks run the risk of relying on the model's parametric knowledge. For example, one of the ICL examples in GPT3 paper is language translation, which clearly heavily depends on pre-trained knowledge of source and target language. We already know that such pre-trained knowledge is impaired in our QA experiments. Synthetic tasks, due to their synthetic nature, are largely free of such issues.
>
> >For ICL results, it seems the performance drops significantly (not gradually) from 70% above, do you have intuition why?
>
> We conjecture that ICL learning ability depends on a small set of weights that act as the in-context gradient descent module [1]. At 70% and above, SparseGPT may eventually decide to remove this small set of weights, causing sudden drop of accuracy.
>
> > As the paper only use some particular pruning methods, do you have any opinion on whether the same findings will hold for other pruning methods? It would be good to have a short discussion on the difference between them.
>
> We believe our findings will generalize to other pruning methods. This is because our findings not only apply to pruning (SparseGPT, Wanda), but also to dense down-scaling. The consistency of our results between sparse and dense down-scaling provides evidence that our observation is fundamentally about model size reduction in general, measured in the number of parameters.
>
> > Section 5 is hard to follow without examples, there are multiple notations without explanation, e.g. K (page 7), x, D=4, N=32, etc
>
> We will make sure to improve our writing and let you know once we finish updating the paper!
>
> > Would be good to release the version of datasets that are used for benchmarking
>
> Yes we will release the datasets we use!

---

> > ### Author Response · Authors · 2023-11-20
> > **Follow-up**
> >
> > We're writing to follow up on a few remaining points:
> >
> > > Although down-scaling and pruning are the main topic of the paper, the technical details on methods used is very limited (even in the Appendix too). If space is an issue, I would suggest to cut down the paper motivation which is repeated multiple times throughout the paper.
> >
> > We included detailed explanation of the pruning algorithm behind SparseGPT and Wanda in Appendix O. We link to this section in our pruning algorithm introduction paragraph in Section Preliminary.
> >
> > > As the paper only use some particular pruning methods, do you have any opinion on whether the same findings will hold for other pruning methods? It would be good to have a short discussion on the difference between them.
> >
> > Apart from our previous response, we've updated our paper to show additional pruning results (Appendix P). These results demonstrate the validity of our conclusion with structured pruning algorithm LLM-Pruner, further strengthening the generality of our conclusion.
> >
> > > Section 5 is hard to follow without examples, there are multiple notations without explanation, e.g. K (page 7), x, D=4, N=32, etc
> >
> > We updated Section 5 to include explicit definitions of these notations. We expect to continue improving the clarity of this section.

---

### Official Review · Reviewer_2TYK · 2023-10-30

**Soundness:** 2 fair
**Presentation:** 3 good
**Contribution:** 2 fair
**Rating:** 6
**Confidence:** 2

**Summary:**

Edit - the authors have addressed concerns within my review and also contrasted the novelty of their work with previous pruning/ICL studies.  I'm increasing my score accordingly.

Continuing off recently proposed LLM pruning methods (i.e., SparseGPT and Wanda), the authors explore the effect of LLM pruning on both parametric knowledge (i.e., knowledge memorized by the model) and knowledge learned via ICL.  Towards this end, the authors explore the effect of pruning for pretrained OPT-13B/30B and LLamA-13B/33B models.  The authors main contributions are experiments evaluating the pruned-sparsity levels of the aforementioned models versus (a) parametric knowledge and (b) ICL knowledge, as measured via several Q&A tests; parametric knowledge is measured using a closed-book Q&A test, while ICL knowledge is measured using an open-book Q&A test where the in-context prompt is contains a counter-factual answer compared to the training data.  Furthermore, the authors also test ICL knowledge by providing in-context information detailing parametric functions and measure the models predictive accuracy in this case.

**Strengths:**

The topic is important and interesting and, at a high-level, the experimental design make sense.  Furthermore, as LLM parameter sizes continue to grow, the question of how new pruning methods (i.e., LLM-Pruner, SparseGPT, and Wanda) affect a model's parametric and ICL knowledge is important.  However, more care is required to ensure model performance is accurately being measured.

**Weaknesses:**

# Evaluation
The major weakness of the paper is the effective testing of LLM parametric and ICL knowledge.  In particular, how do the authors verify that the in-context evidence contradicts a fact present in training data?  Although LLama was trained on
publicly available data, it is not a simple matter to verify that the answers in the Q&A datasets align or misalign with the massive dataset used to train LLamA (note that, the dataset itself was never released, so all public datasets would have to be evaluated in their entirety for verification).  In this case, it is not clear how Verifying that "answers do not exist in the pre-training corpus" is possible.  While the authors discuss previous work which has explored LLMs' abilities to override in-memory/parametric knowledge, such works set up measurement guardrails to do so through fine-tuning, e.g.:
- "DissentQA. Neeman et al. (2022) constructed the DissentQA dataset from the NaturalQuestions dataset. It contains pairs of
questions and evidence for a made-up answer that is different from the factual one. It assesses
whether the model can override its memory formed during pre-training with new context."
"Given that the anticipated made-up answers are randomized and
different from the factual ones, the model cannot depend on memorization from its pre-training
data to generate responses. This evaluation framework rigorously assesses the model’s ability to
override its pre-training data with new, context-specific information." <- Neeman et al. (2022) fine-tune their evaluated T5 models, thus ensuring that the parameterized answers are learned and relevant questions and answers are actually counter-factual (relatedly, gold passages are considered in Longpre et al. (2021)). This work does not fine-tune the evaluated LLMs, thus relating to the earlier criticism on the validity of the presented results.

Furthermore, the exact measurement of accuracy used in the paper is potentially incorrect and too conservative for recently release instruction-tuned LLMs like LLaMA and OPT.  From the text:
- "Answers are the model’s prompt completions produced by greedy decoding. We report the
percentage of answers that exactly match ground truth." <- Two important remarks: greedy is known to be extremely suboptimal for recent
instruction-tuned LLMs, and an exact match is not necessarily a fair metric.  Such chat models are known to be extremely wordy, so if the model produces some lead up text followed by the correct answer, this metric discounts such a correct response.  For the former, it makes sense as a fair, reproducible benchmark across different sparsity percentages per model (e.g., nucleus sampling would produce differing results between runs), please include in the text why greedy is used.  However, note that the latter is an extremely important problem which biases all related results.

# Claims
Several claims require revision or further discussion.  In general, wrt to key contributions, it is necessary to discuss how the presented methodology differs from previous work.  E.g., the SparseGPT paper itself reports zero-shot performance for different datasets at different sparsity levels (which effectively tests parametric knowledge), how does the presented benchmark differ from this?  Why does the presented work differ in conclusions wrt parametric knowledge compared to the SparseGPT paper, i.e., SparseGPT showed high sparsity while retraining zero-shot performance.  Why is this not the case in the presented work?  These types of questions, and their ensuing answers/justifications, require significant discussion.  More examples from the paper:
- "From work on image classification, however, we know that down-scaling neural networks affects more than just top-line metrics or task accuracy. Pruning, for example, can introduce biases (Hooker et al., 2019) or disproportionate affects on certain subsets of the data (Jin et al., 2022)." <- This claim is too strong, it makes it seem as though it is a certainty that such effects occur given down-scaling.  However, a significant amount of work has shown that pruning is an effective tool for vision models.
- "It is difficult to assess these abilities in isolation, as a standard downstream task needs to process the
information provided in context as well as access the information stored in weights." <- Please contrast related work which has previously explored zero-to-many shot ICL performance (across different target applications); see the following for an extensive overview:
Dong et al, "A Survey on In-context Learning", https://arxiv.org/pdf/2301.00234.pdf
- "Improve inference efficiency. Our work reveals that scaling down model size alone has little impact
on tasks demanding processing information in the LLM’s context. Practitioners may thus use our
findings to identify scenarios where decisions could be routed to a smaller model instead of a larger
one without hurting task performance (Chen et al., 2023; Dohan et al., 2022)." <- The latter work already explores how the parameter size affects performance.  In particular, the Wanda paper already tackles the question of how pruning affects ICL performance (and compares to SparseGPT)
- "Our work differs from prior scaling studies in two ways: while prior work (Kaplan et al., 2020b) studies
joint scaling of both pre-training corpus size and model size, we focus on scaling model size alone.
Furthermore, instead of measuring task performance, we focus on foundational capabilities of LLMs–fact recall and ICL. These capabilities drive the success for many real world applications of LLMs" <- This is wrong for a number of reasons.  Firstly, "we focus on scaling model size along" is not a valid contribution, as this would, by definition, be provided in the study of "joint scaling of both pre-training corpus size and model size."  Secondly, the work of Kaplan does not study pruning, but rather LLM model size->training->resulting performance.  It is necessary to demarcate the difference between these two paradigms.
-"In-weight versus in-context learning" <- Please explain how the considered work differs from Longpre et al 2022, which extensively explores In-weight versus in-context learning.
- "the versatility of LLMs calls for a different approach to assessing pruned models. Our work begins to fill this gap, proposing to evaluate pruning’s effect on fact recall and ICL." <- As previously mentioned, what the authors define as fact recall is equivalent to the task of zero-shot question answering; the effect of pruning on various tasks has been explored, e.g., within the papers of the pruners specifically used within this work (SparseGPT and Wanda), as well as in the LLM-Pruner paper.  Furthermore, the effect of pruning an LLM to various sparsity levels on ICL was extensively explored in the Wanda paper.  Please revise your contributions, and position them within the context of previous works.
-" In all the above settings, as a simple point of comparison, we measure the effect of downscaling on perplexity" <- Please note in the paper that this was previously considered in both the SparseGPT and Wanda papers.
-"  we focus on foundational capabilities of LLMs – fact recall and ICL" <- Fact recall is zero-shot Q&A, which may be thought of as a specific task.  Please adjust this claim.

# Presentation
Overall, the writing and presentation of the discussed work could be significantly improved.  E.g.:
- "removing more than 30% of weights leads to significant (> 5%, relative) accuracy degradation on fact recall related tasks (Figure 1, left). Fact recall suffers similarly from dense down-scaling." <- Please have some text which segways from the first paragraph (page 2) to the list of 3 bold-faced items.  The intro currently reads like a collection of text/paragraphs which do not blend into one another.  E.g., combine all the bold-faced-starting paragraph in page 2 into a single paragraph, which: -States the paper shows the following dichotomy wrt pruning LLMs.  For fact recall/parametricknowldge, minimal pruning significantly degrades performance.  [insert your bold-faced-starting text here] In stark contrast, large-scale  pruning does not significantly degrade ICL performance. [insert your second bold-faced-starting text here]. [insert your third bold-faced-starting text here]
- Same comment for italicized-starting-text, which proceed bold-starting-text; please segway the various paragraphs together.  It is very difficult for a reader to understand the point that is trying to be made when sentences exist independently.

**Questions:**

-Why did the authors not consider the GINC dataset for ICL, from Xie et al's "An Explanation of In-context Learning as Implicit Bayesian Inference?"

-"From the OPT family, we evaluate the two largest models that fit in our hardware
setup" <- Please state the hardware setup

-In Table 1, please define what is meant by "Context Type"

---

> ### Author Response · Authors · 2023-11-17
> **Response - Evaluation**
>
> We appreciate the efforts that go into providing us with so much feedback! We strive to address them fully, and we look forward to hearing back!
>
> > The major weakness of the paper is the effective testing of LLM parametric and ICL knowledge. In particular, how do the authors verify that the in-context evidence contradicts a fact present in training data? Although LLama was trained on publicly available data, it is not a simple matter to verify that the answers in the Q&A datasets align or misalign with the massive dataset used to train LLamA (note that, the dataset itself was never released, so all public datasets would have to be evaluated in their entirety for verification). In this case, it is not clear how Verifying that "answers do not exist in the pre-training corpus" is possible. While the authors discuss previous work which has explored LLMs' abilities to override in-memory/parametric knowledge, such works set up measurement guardrails to do so through fine-tuning
>
> We agree with your point and took actions to address your concern. It is indeed nearly impossible to verify that counter-factual answers do not exist in pre-training corpus. We amended the phrasing of our experiment design to avoid making absolute statement about dataset contamination. Furthermore, we quantify the extent to which they exist in the pre-training corpus, by running the counterfactual QA evaluation using dense models without the counterfactual context – we count the model’s answer as correct if and only if it matches the counterfactual answer. If the training dataset of LLaMA is contaminated with the counterfactual question/answers, we should expect to see reasonably high accuracy under this setup. Instead, we observe near-zero accuracy for both LLaMA-13 and LLaMA-33B model (0.1% for both LLaMA-13B and LLaMA-33B).
>
> >Furthermore, the exact measurement of accuracy used in the paper is potentially incorrect and too conservative for recently release instruction-tuned LLMs like LLaMA and OPT. From the text:
> "Answers are the model’s prompt completions produced by greedy decoding. We report the percentage of answers that exactly match ground truth." <- Two important remarks: greedy is known to be extremely suboptimal for recent instruction-tuned LLMs, and an exact match is not necessarily a fair metric. Such chat models are known to be extremely wordy, so if the model produces some lead up text followed by the correct answer, this metric discounts such a correct response. For the former, it makes sense as a fair, reproducible benchmark across different sparsity percentages per model (e.g., nucleus sampling would produce differing results between runs), please include in the text why greedy is used. However, note that the latter is an extremely important problem which biases all related results.
>
> To begin with, both LLaMA[4] and Pythia[5] models (which we study) use greedy decoding and exact match during evaluation. So our experimental design is standard as is. Furthermore, we do not use instruction-tuned models, so verbosity should not be a concern either. Nevertheless, we collected extra results to address your concern about decoding and wordiness; and show that our conclusion holds up to additional scrutiny. We re-run key subset of our experiments using a more advanced sampling scheme and more tolerant matching criteria according to your proposal. Specifically, we repeat our LLaMA-13B and LLaMA-33B experiments on TriviaQA benchmark with and without context. We use beam search with beam size of 3 (some paper in the past, e.g., OPT[6], GPT3[7] used beam search during evaluation). To get an answer from the model, we generate 32 tokens from the model, and check whether the answer appears anywhere within the generated tokens. The average number of tokens for TriviaQA answer is 4.7, so a model can generate extra tokens irrelevant to the answer whilst still correctly answering the question. Here’re our results:
> - Our conclusion still holds in this setting: using our current experimental design, averaging over two LLaMA models, one can prune 30% and 40% weights on Closebook and Openbook TriviaQA task, respectively. Using beam search, one can prune 30% and 50% weights on Closebook and Openbook TriviaQA tasks, respectively. The ability to retrieve answers from provided context remains more resilient to pruning than the ability to recall information learnt during pre-training.
> - Additionally, we observe a noticeable accuracy boost likely due to better decoding strategy using beam search (about 10-20% accuracy improvement). This is expected, as beam search with beam size of 3 costs ~3x the compute of greedy decoding.
> - Please see full pruning curves and comparison with greedy decoding in Appendix L of our newly updated paper draft.

---

> ### Author Response · Authors · 2023-11-17
> **Response - Claims (1/2)**
>
> > "From work on image classification, however, we know that down-scaling neural networks affects more than just top-line metrics or task accuracy. Pruning, for example, can introduce biases (Hooker et al., 2019) or disproportionate affects on certain subsets of the data (Jin et al., 2022)." <- This claim is too strong, it makes it seem as though it is a certainty that such effects occur given down-scaling. However, a significant amount of work has shown that pruning is an effective tool for vision models.
>
> We agree. Our use of the word “can” intends to convey the uncertainty. If you think this is insufficient, we are happy to be more precise about the precondition of the said phenomenon – “Pruning to very high sparsity, for example, can introduce biases…”
>
> > "It is difficult to assess these abilities in isolation, as a standard downstream task needs to process the information provided in context as well as access the information stored in weights." <- Please contrast related work which has previously explored zero-to-many shot ICL performance (across different target applications); see the following for an extensive overview: Dong et al, "A Survey on In-context Learning", https://arxiv.org/pdf/2301.00234.pdf
>
> Thank you for sharing the survey on ICL. Our definition of ICL is consistent with the one provided in the survey. The survey summarizes work on various aspects of ICL (algorithmic mechanisms behind it, evaluation of ICL, benchmarks, improving ICL, ICL in smaller models via distillation, etc.), but does not mention any work attempting to disentangle ICL and fact recall, or evaluating how pruning affects ICL, thus providing further evidence on the novelty of our contribution.
>
> > "Improve inference efficiency. Our work reveals that scaling down model size alone has little impact on tasks demanding processing information in the LLM’s context. Practitioners may thus use our findings to identify scenarios where decisions could be routed to a smaller model instead of a larger one without hurting task performance (Chen et al., 2023; Dohan et al., 2022)." <- The latter work already explores how the parameter size affects performance. In particular, the Wanda paper already tackles the question of how pruning affects ICL performance (and compares to SparseGPT)
>
> Please also refer to our global response for a discussion of our contribution within the context of existing pruning work.
>
> FrugalGPT[8] (Chen et al.,), SparseGPT[1] and Wanda[2] indeed all investigated the task performance-size/cost trade-off of LLMs. However, when a new task emerges outside of those examined by these prior studies, how can a practitioner know if it is a good idea to route it to a smaller model? Our work provides insight in this scenario: practitioners can assess how much the said task relies on parametric knowledge versus context processing capability. The more a task relies on context processing, the more likely such a task can be routed to a smaller model.
>
> > "Our work differs from prior scaling studies in two ways: while prior work (Kaplan et al., 2020b) studies joint scaling of both pre-training corpus size and model size, we focus on scaling model size alone. Furthermore, instead of measuring task performance, we focus on foundational capabilities of LLMs–fact recall and ICL. These capabilities drive the success for many real world applications of LLMs" <- This is wrong for a number of reasons. Firstly, "we focus on scaling model size along" is not a valid contribution, as this would, by definition, be provided in the study of "joint scaling of both pre-training corpus size and model size."
>
> We use this statement to highlight a difference in methodological choice – we want to be upfront about the more focused and thus narrower scope of our study. To be clear, we use this statement to disambiguate, not to present a contribution.
>
> > Secondly, the work of Kaplan does not study pruning, but rather LLM model size->training->resulting performance. It is necessary to demarcate the difference between these two paradigms.
>
> We agree. The difference is important – 1). In addition to dense scaling (Sec 6), which follows the paradigm of Kaplan et al, we also studied pruning. 2). Instead of perplexity, we assess LLMs on its ability to recall information from pretraining vs process contextual information. Both differences are also our key contributions.

---

> ### Author Response · Authors · 2023-11-17
> **Response - Claims - 2/2**
>
> > "In-weight versus in-context learning" <- Please explain how the considered work differs from Longpre et al 2022, which extensively explores In-weight versus in-context learning.
>
> This paper focuses on constructing experiments that contradict existing knowledge in a variety of ways, including changing the answer to an alias, changing the type of the answer. Alongside DisentQA they provide key methodological foundations to our work (specifically in the counterfactual QA setup). Our work differs from this in that 1). Our work studies pruning, and 2). Our work investigates context-processing abilities beyond answering counter-factual questions – we also construct in-context learning tasks to learn parametric functions to stress test a model’s ability to process information in context.
>
> > "the versatility of LLMs calls for a different approach to assessing pruned models. Our work begins to fill this gap, proposing to evaluate pruning’s effect on fact recall and ICL." <- As previously mentioned, what the authors define as fact recall is equivalent to the task of zero-shot question answering; the effect of pruning on various tasks has been explored, e.g., within the papers of the pruners specifically used within this work (SparseGPT and Wanda), as well as in the LLM-Pruner paper. Furthermore, the effect of pruning an LLM to various sparsity levels on ICL was extensively explored in the Wanda paper. Please revise your contributions, and position them within the context of previous works.
>
> > we focus on foundational capabilities of LLMs – fact recall and ICL" <- Fact recall is zero-shot Q&A, which may be thought of as a specific task. Please adjust this claim.
>
> We emphasize that fact recall is not equivalent to zero-shot closebook question answering (correct us if we’re wrong, by zero-shot QA we believe you mean zero-shot **closebook** QA). The model may rely on the ability to recall parametric knowledge even in openbook QA settings. Hence we argue that no single task exclusively test the model’s ability to recall information from pretraining versus its ability to process information in context. Hence a primary contribution of ours is to disentangle the assessment of both abilities to the best of our abilities. Please refer to our global response for the discussion of our contribution in the context of prior pruning work.
>
> Nevertheless, we are happy to better acknowledge the work done by prior pruning research, and revised our phrasing to the following: … approach to assessing pruned models. Prior work does a commendable job to assess model size and task accuracy trade-off. Our work continues to expand our toolkits for empirical assessment, proposing to evaluate pruning’s effect on fact recall and ICL.
>
> > " In all the above settings, as a simple point of comparison, we measure the effect of downscaling on perplexity" <- Please note in the paper that this was previously considered in both the SparseGPT and Wanda papers. -"
>
> Agree. Revised to credit this to prior pruning papers.

---

> > ### Author Response · Authors · 2023-11-17
> > **Response -- Presentation & Questions**
> >
> > > "removing more than 30% of weights leads to significant (> 5%, relative) accuracy degradation on fact recall related tasks (Figure 1, left). Fact recall suffers similarly from dense down-scaling." <- Please have some text which segways from the first paragraph (page 2) to the list of 3 bold-faced items. The intro currently reads like a collection of text/paragraphs which do not blend into one another. ...
> >
> > Thank you for your suggestions! We applied your suggested comments; since this involves substantial change in our intro text, we expect to continue updating and improving it during the discussion period.
> >
> > > Same comment for italicized-starting-text, which proceed bold-starting-text; please segway the various paragraphs together. It is very difficult for a reader to understand the point that is trying to be made when sentences exist independently.
> >
> > Unlike the previous point, these individual items are indeed discussing largely independent topics. We feel like they may be better left as independent paragraphs.
> >
> > >-Why did the authors not consider the GINC dataset for ICL, from Xie et al's "An Explanation of In-context Learning as Implicit Bayesian Inference?"
> >
> > GINC dataset requires pre-training on the synthetically generated pre-training dataset for the evaluation task to make sense. The task is purely synthetic, a model pre-trained on natural language corpus cannot not solve it. Since our focus is on evaluating pre-trained LLMs, this task does not appear to suit our needs. Please see the eval dataset sample here [9].
> >
> > > -"From the OPT family, we evaluate the two largest models that fit in our hardware setup" <- Please state the hardware setup
> >
> > Our hardware setup is available in appendix J. All our evaluations are performed on TPU v3-8; each core of TPU v3 has 16GB HBM memory. We used batch size 1 and additional tricks such as CPU-offloading to fit larger models inside TPUs.
> >
> > > -In Table 1, please define what is meant by "Context Type"
> >
> > It means what is the type of supporting evidence we insert into the model’s context to help the model solve the task. Empty means no evidence provided, and is used in closebook QA evaluation. Factual means we provide a paragraph of supporting evidence containing the factual answer to the question, and is used in openbook QA evaluation. Synthetic means we provide a paragraph of evidence containing the counterfactual answer to the question, and is used in counterfactual QA evaluation. Exemplary means we provide in-context examples to guide models to learn parametric functions in-context.
> >
> > [1] SparseGPT: Massive Language Models Can be Accurately Pruned in One-Shot, Frantar et al., ICML 2023.
> >
> > [2] A Simple and Effective Pruning Approach for Large Language Models, Sun et al., arXiv 2023.
> >
> > [3] LLM-Pruner: On the Structural Pruning of Large Language Models, Ma et al., NeurIPS 2023.
> >
> > [4] LLaMA: Open and Efficient Foundation Language Models, Touvron et al., arXiv 2023.
> >
> > [5] Pythia eval code: https://github.com/EleutherAI/lm-evaluation-harness/blob/master/lm_eval/tasks/triviaqa.py
> >
> > [6] OPT: Open Pre-trained Transformer Language Models, Zhang et al., arXiv 2022.
> >
> > [7] Language Models are Few-Shot Learners, Brown et al., NeurIPS 2020.
> >
> > [8] FrugalGPT: How to Use Large Language Models While Reducing Cost and Improving Performance, Chen et al., arXiv, 2023.
> >
> > [9] GINC sample eval data https://raw.githubusercontent.com/p-lambda/incontext-learning/main/data/GINC_trans0.1_start10.0_nsymbols50_nvalues10_nslots10_vic0.9_nhmms10/id_prompts_randomsample_3.json
> >
> > [10] A Survey on In-context Learning, Dong et al., arXiv 2022.
> >
> > [11] OpenICL: An Open-Source Framework for In-context Learning, Wu et al., arXiv 2023.
> >
> > [12] Entity-Based Knowledge Conflicts in Question Answering, Longpre et al., ACL 2021.
> >
> > [13] DisentQA: Disentangling Parametric and Contextual Knowledge with Counterfactual Question Answering, Neeman et al., arXiv 2022.

---

> > > ### Author Response · Authors · 2023-11-21
> > > **Reminder**
> > >
> > > Dear Reviewer 2YTK:
> > >
> > > We are looking forward to hearing whether our response have fully addressed your concerns!
> > >
> > > Best wishes.

---

> > > ### Comment · Reviewer_2TYK · 2023-11-21
> > > **Reply to rebuttal**
> > >
> > > I've read all reviews and responses.  I thank the authors for their extensive answer to all my questions/remarks and increased my score accordingly.

---

### Official Review · Reviewer_xD3m · 2023-11-01

**Soundness:** 3 good
**Presentation:** 3 good
**Contribution:** 2 fair
**Rating:** 6
**Confidence:** 4

**Summary:**

Pruning parameters from large language models can affect aspects of model performance differently. The authors strive to characterize these effects by separating fact recall from in context learning. They explore the relative impact of pruning on several different tasks using several base models and multiple pruning techniques. Overall they find that even moderate pruning can degrade fact recall settings, here in-context learning seems more robust.

**Strengths:**

The settings for evaluating fact recall and in context learning seem useful in general.

Multiple settings for pruning to push for more robust result interpretations

Multiple model families were used in evaluation.

A range of tasks were presented.

**Weaknesses:**

Fact Recall and In Context Learning are some reasonable aspects, but the authors could have considered more. Detailed Instruction Following, and Heavy Reasoning feel like other key aspects, as well as the ability to learn from Few Shot inline. I would have loved to see some more details.

Are all In Context Learning tasks equally difficult? Could a few more gradations be helpful here?

Are there a few more settings that one could use for evaluating model performance? The set of tasks seems rather small.

I'm assuming that pruning is primarily used to increase inference speed, right? If that's the case, I'd like to see tradeoffs between accuracy and inference speed be presented here.

**Questions:**

It seems that Dense Pruning of 30B -> ~13B underperforms the unpruned 13B param model, right? I'd love to see more discussion here about what is going on there.

---

> ### Author Response · Authors · 2023-11-17
> **Response**
>
> Thank you for your helpful feedback! Here's our response to your comments.
>
> > Fact Recall and In Context Learning are some reasonable aspects, but the authors could have considered more. Detailed Instruction Following, and Heavy Reasoning feel like other key aspects, as well as the ability to learn from Few Shot inline. I would have loved to see some more details.
> > Are there a few more settings that one could use for evaluating model performance? The set of tasks seems rather small.
>
> Please see the additional task evaluation section of our global response. Specifically, we present additional ICL task evaluation to learn algorithmic solutions to problems in context. The solution thus requires some algorithmic reasoning. For the ability to learn from few-shots inline, we emphasize that our ICL task evaluation setup examines exactly this ability. Model learns to perform novel tasks based on few-shot examples we provide in-context.
>
> As for detailed instruction following, since all six models we examine are pre-trained models without instruction-tuning, we feel like this may be out of scope for our current work.
>
> > Are all In Context Learning tasks equally difficult? Could a few more gradations be helpful here?
>
> In Appendix E, we varied the difficulty of in-context learning tasks by changing the input dimensionality, and observed that ICL tasks nevertheless remain highly resilient to pruning. This observation corroborates our main conclusion.
>
> > I'm assuming that pruning is primarily used to increase inference speed, right? If that's the case, I'd like to see tradeoffs between accuracy and inference speed be presented here.
>
> It is true that the ultimate goal of pruning is to increase inference speed. However, pruning studies usually appear before the relevant hardware support is available. For example, seminal work done by Han et al. showed the first promising results of applying pruning techniques to deep neural network models in 2015. However it is not until 2020 that nVidia rolled out the first GPU with hardware support for sparsity. Our work focuses on unstructured sparsity, which is not supported natively by all current generations of GPUs and TPUs; nonetheless, we believe our results will facilitate the real-world adoption of pruning techniques like SparseGPT and Wanda.

---

### Official Review · Reviewer_ecKe · 2023-11-03

**Soundness:** 2 fair
**Presentation:** 3 good
**Contribution:** 2 fair
**Rating:** 6
**Confidence:** 4

**Summary:**

The authors study the effects of weight pruning, a popular technique for reducing model size, on the two core capabilities of LLMs: (a) recalling facts presented during pre-training and (b) processing information presented in context. They find that existing pruning techniques affect these two abilities of LLMs quite differently. The paper presents a detailed analysis of the experimental results, which show that the effects of down-scaling LLMs depend on the specific pruning technique used. The authors conclude that there is a trade-off between model size and performance, and that the optimal model size depends on the specific task and dataset.

**Strengths:**

- The paper investigates the impact of down-scaling large language models on their capabilities, which is an important topic in the field of natural language processing.
- The authors provide a detailed analysis of the experimental results, which can help researchers and practitioners better understand the trade-offs between model size and performance.
- The paper provides insights into the development of more efficient language models, which are becoming increasingly important for a wide range of natural language processing tasks.

**Weaknesses:**

- The paper is empirical in nature, and the authors acknowledge that their observations may not generalize to the full spectrum of tasks and large language models.
- The study focuses on evaluating two pruning algorithms that are unstructured pruning, evaluation on structured pruning methods are expected.
- The study could include other types of tasks, like NLI, classification, summarization, to make the study more solid.

**Questions:**

How the structured pruning methods, e.g., LLM-Pruner, performs on these two LLM capabilities?

---

> ### Author Response · Authors · 2023-11-17
> **Response**
>
> Thank you for your helpful feedback!
>
> > The study focuses on evaluating two pruning algorithms that are unstructured pruning, evaluation on structured pruning methods are expected.
> > How the structured pruning methods, e.g., LLM-Pruner, performs on these two LLM capabilities?
>
> We are working on providing some structured pruning results. We will get back to you once results become available.
>
> > The study could include other types of tasks, like NLI, classification, summarization, to make the study more solid.
>
> Please see additional task evaluation in our global rebuttal response. We present 4 additional task evaluations including lambada, translation and additional ICL classification tasks.

---

> > ### Author Response · Authors · 2023-11-20
> > **Structured pruning update**
> >
> > We're writing to follow up on structured pruning results:
> >
> > Please refer to our Appendix P. Structured Pruning Results, our conclusion remains valid with structured pruning algorithms LLM-pruner. Structured pruning results demonstrate that in-context learning of parametric functions is more robust to pruning compared to Openbook QA. Openbook QA itself shows more robustness to pruning than Closebook QA. This is consistent with the conclusion from our main paper.

---

### Author Response · Authors · 2023-11-17
**Global Rebuttal Response**

We appreciate all reviewers for the insightful comments! In this global response, we highlight some of our efforts to address reviewer concerns.

- **Additional tasks**. Many reviewers (ecKe, xD3m, B31e) would like to see additional task evaluations. We therefore present 4 additional task evaluations below. The results are consistent with the conclusion of our paper. Details are available in the Additional task evaluation section below.
- **Experiment design**. Reviewer 2TYK questioned our evaluation methodology using greedy decoding and exact match to generate and check model answers. We emphasize that this is a standard evaluation protocol consistent with LLaMA/Pythia[4, 5]. Nonetheless, we perform additional experiments with beam search and a more tolerant answer-matching method. Our conclusion remains unchanged. More details available in Fig.15 of our updated Appendix L., and in our response to 2TYK.
- **Contribution**. Reviewer 2TYK asked for more discussion of our contribution in the context of prior work that also performed task-level LLM evaluations. We emphasize that  no single task solely tests a model's ability for recalling pre-training information versus processing contextual information (see 3.1 Evaluation in our paper or the pruning section of Discussion of contribution below for explanation). Thus, a distinct contribution of ours is developing benchmarks that maximally disentangle the assessment of these two complementary abilities. We present more detailed discussion in the context of three lines of related work below (Discussion of contribution).

## Additional task evaluation. ##

We present several additional tasks including translation, lambada (an ICL task with natural language output) and algorithmic ICL tasks (classification tasks requiring algorithmic reasoning). Our observations are below:

- **Lambada task [Appendix M, Fig. 16].** Lambada is a challenging ICL task that involves natural language generation and long-range reasoning. The dataset evaluates the model's ability to complete a paragraph with a single word. The dataset design ensures that "models cannot simply rely on local context, but must be able to keep track of information in the broader discourse" [10]. We find that our observation that ICL ability survives to high sparsity similarly holds on this dataset, where the accuracy drop is within 5% of the dense model even at 60% sparsity.
- **Translation task [Appendix M, Fig. 17].** As argued in our paper, translation tasks require both in-context processing and knowledge recall to perform well: it needs to adhere strictly to the content placed in its context in its translation whilst leveraging its pretrained knowledge of the source/destination language to perform well. As expected, we see middling robustness to pruning compared with tasks that are purely fact recall/ICL– we may prune both LLaMA 13B and 30B to 40% sparsity without losing more than 5% the BLEU score of the dense model.
- **Algorithmic ICL tasks [Appendix N, Fig. 18, 19].** In the newly updated Appendix N, we illustrate two additional ICL tasks that require learning algorithmic solutions to a synthetic task: ICL ability is consistently resilient to pruning to 60-70% sparsity.

## Discussion of contribution. ##
Here’s a short discussion of our contribution in relation to several lines of related work:
- **Pruning(e.g., [1], [2], [3]):** all prior pruning work did a commendable job at evaluating pruning techniques on a variety of tasks beyond perplexity, some variants of benchmarks in this paper (BoolQ, OpenBook QA in [2, 3] for example) appeared in those papers, too. However, benchmarks like OpenBook QA do not exclusively test a model's context processing ability: the presence of context does not prevent the model from relying on parametric knowledge and completely ignoring the context. This limitation has led us to include several unconventional benchmarks in evaluation of pruned models, such as counterfactual QA and in-context learning of parametric functions, which were not part of existing pruning studies. Our unique experimental design has yielded equally unique findings: it is possible to prune up to 70% on tasks that predominantly require context processing, like counterfactual QA and in-context learning for parametric functions. We strongly believe that the community would benefit from the knowledge of the surprising effectiveness of pruning algorithms in these scenarios.

- **ICL survey/benchmark (e.g., [6, 7]):** to the best of our knowledge, no prior ICL studies have attempted to disentangle fact recall from ICL.

- **Parametric/contextual knowledge conflict (e.g., [8, 9]):** these papers provide the methodological foundation to our work; however, they do not study pruning, nor do they investigate the context processing ability of LLMs beyond question-answering setup, such as learning parametric functions in context, which we evaluate.

---

> ### Author Response · Authors · 2023-11-17
> **Global Rebuttal Response References**
>
> [1] SparseGPT: Massive Language Models Can be Accurately Pruned in One-Shot, Frantar et al., ICML 2023.
>
> [2] A Simple and Effective Pruning Approach for Large Language Models, Sun et al., arXiv 2023.
>
> [3] LLM-Pruner: On the Structural Pruning of Large Language Models, Ma et al., NeurIPS 2023.
>
> [4] LLaMA: Open and Efficient Foundation Language Models, Touvron et al., arXiv 2023.
>
> [5] Pythia eval code: https://github.com/EleutherAI/lm-evaluation-harness/blob/master/lm_eval/tasks/triviaqa.py
> https://raw.githubusercontent.com/p-lambda/incontext-learning/main/data/GINC_trans0.1_start10.0_nsymbols50_nvalues10_nslots10_vic0.9_nhmms10/id_prompts_randomsample_3.json
>
> [6] A Survey on In-context Learning, Dong et al., arXiv 2022.
>
> [7] OpenICL: An Open-Source Framework for In-context Learning, Wu et al., arXiv 2023.
>
> [8] Entity-Based Knowledge Conflicts in Question Answering, Longpre et al., ACL 2021.
>
> [9] DisentQA: Disentangling Parametric and Contextual Knowledge with Counterfactual Question Answering, Neeman et al., arXiv 2022.
>
> [10] The LAMBADA dataset: Word prediction requiring a broad discourse context., Paperno et al., ACL 2016.

---

### Meta-Review · Area_Chair_U9LU · 2023-12-11

**Metareview:**

This paper studies the impact of weight pruning on the capabilities of LLMs, focusing on factuality and reasoning. The paper is well positioned with regard to relevant prior work and makes clear its novel contributions, which center around disentangling factual recall and in-context learning. As such, future work can use some of these findings to decide how much of the LLM to prune based on their task's demands. The reviewers raised concerns about the diversity and number of tasks evaluated, which the authors addressed during the response period by adding four more tasks (including LAMBADA and MT). Overall, the paper makes a nice (albeit somewhat incremental) contribution to the study of weight pruning within LLMs and deserves consideration of acceptance into ICLR.

**Justification For Why Not Higher Score:**

The paper is empirical in nature, which leaves natural open questions (brought up by reviewers) on the generalizability of these findings to other pruning schemes and downstream tasks.

**Justification For Why Not Lower Score:**

The paper's findings are interesting and the authors have conducted a comprehensive set of experiments. I think the paper overall will be useful to anyone working on pruning (as perhaps an alternate evaluation compared to more task-specific measures vs. speed).

---

### Decision · Program_Chairs · 2024-01-16

Accept (poster)